# Automated cryo-EM structure refinement using correlation-driven molecular dynamics

Maxim Igaev*, Carsten Kutzner, Lars V Bock, Andrea C Vaiana*, Helmut Grubmüller*

Department of Theoretical and Computational Biophysics, Max Planck Institute for Biophysical Chemistry, Göttingen, Germany

**Abstract** We present a correlation-driven molecular dynamics (CDMD) method for automated refinement of atomistic models into cryo-electron microscopy (cryo-EM) maps at resolutions ranging from near-atomic to subnanometer. It utilizes a chemically accurate force field and thermodynamic sampling to improve the real-space correlation between the modeled structure and the cryo-EM map. Our framework employs a gradual increase in resolution and map-model agreement as well as simulated annealing, and allows fully automated refinement without manual intervention or any additional rotamer- and backbone-specific restraints. Using multiple challenging systems covering a wide range of map resolutions, system sizes, starting model geometries and distances from the target state, we assess the quality of generated models in terms of both model accuracy and potential of overfitting. To provide an objective comparison, we apply several well-established methods across all examples and demonstrate that CDMD performs best in most cases.
DOI: https://doi.org/10.7554/eLife.43542.001

**\*For correspondence:**
migaev@mpibpc.mpg.de (MI);
Andrea.Vaiana@mpibpc.mpg.de (ACV);
hgrubmu@gwdg.de (HG)

**Competing interests:** The authors declare that no competing interests exist.

## Introduction

State-of-the-art cryo-electron microscopy (cryo-EM) allows biomolecules to be resolved in different functional states and at near-atomic resolution previously achieved only by X-ray crystallography (*Zhang and Nogales, 2015*). A new generation of electron detectors and the development of sample motion correction algorithms have been the major contributors to this rapid progress (*Bai et al., 2015*). Cryo-EM maps determined to resolutions better than 4 Å have become standard with several examples claimed to break the 2Å-resolution barrier (*Frank, 2017*). Yet, benefiting from the recent advances in image classification (*Frank and Ourmazd, 2016*; *Nakane et al., 2018*), cryo-EM now provides ensembles of low- to high-resolution reconstructions describing differently populated conformational states of vitrified molecular complexes. However, methods for deriving accurate atomistic models from cryo-EM maps lag behind this *resolution revolution* (*Saibil, 2017*). The increasing amount of molecular detail requires the development of new methodologies and software to accurately and timely interpret experimental densities.

For de novo building of polypeptide/nucleotide chains many methods are available (*Emsley and Cowtan, 2004*; *Langer et al., 2008*; *Terwilliger et al., 2008*; *Adams et al., 2010*; *Burnley et al., 2017*), including tools for subsequent refinement of hand-built or homology models into densities (*Chen et al., 2003*; *Fabiola and Chapman, 2005*; *Schröder et al., 2007*; *Topf et al., 2008*; *Brown et al., 2015*; *Lopéz-Blanco and Chacón, 2013*; *Wu et al., 2013*; *DiMaio et al., 2015*; *Wang et al., 2016a*; *Kovalevskiy et al., 2018*). In these methods, stereochemical properties and local electrostatics are not fully enforced, nor is a proper description of large-scale concerted motions guaranteed. When applied to maps with medium (local) resolutions or lower, they tend to

produce errors such as atomic clashes, improper bond lengths and angles, and poor agreement between the density and the corresponding model (*Hooft et al., 1996*; *Neumann et al., 2018*). As a result, multiple rotamer, backbone and secondary structure restraints, manual intervention on the system during refinement, and subsequent re-refinement are often required to preserve stereochemical plausibility and avoid overfitting.

As an alternative, several molecular dynamics (MD)-based refinement methods have been developed (*Brünger et al., 1987*; *Brunger and Adams, 2002*; *Trabuco et al., 2008*; *Orzechowski and Tama, 2008*; *Fenn et al., 2011*; *McGreevy et al., 2016*; *Miyashita et al., 2017*; *Wang et al., 2018*). A more comprehensive overview of how MD simulations can be used to assist structure refinement is given elsewhere (*Kirmizialtin et al., 2015*). Here, the refinement rests on a chemically accurate force field and thermodynamic sampling of conformations in addition to guiding the model into the density, as opposed to minimizing a simple target function. These MD-based methods have markedly higher computational requirements which are usually met by today's efficient parallel and GPU implementations. High-resolution maps, however, pose a challenge to MD-based refinement (*McGreevy et al., 2016*). Frequently, parts of the system get trapped due to inefficient conformational sampling in rugged density regions commonly present in such maps. Despite recent improvements (*McGreevy et al., 2016*; *Miyashita et al., 2017*), the obtained models still strongly depend on local map resolution and the choice of a particular refinement scheme or force field. Also, systematic, force-field driven deviations in model geometry have been reported (*Wang et al., 2018*).

It is therefore fair to say that no available cryo-EM refinement method simultaneously meets the three challenges posed by the current resolution revolution: resolution-independent density fitting, stereochemical accuracy, and automation. Here, we have developed and implemented such an automated yet accurate method for structure refinement and validation, denoted correlation-driven molecular dynamics (CDMD), by combining a previously published methodology for low-resolution maps (*Orzechowski and Tama, 2008*) with continuously adaptive resolution and simulated annealing (*Brünger et al., 1987*; *Brunger and Adams, 2002*). In all the presented application examples, we challenge CDMD to refine distant starting models of diverse quality against cryo-EM densities at various resolutions (2.6–7 Å). Performance is assessed using commonly defined measures such as the radius of convergence, model geometry and overfitting. A full analysis of the results and a discussion of the practical conclusions in the general context of structure refinement are presented.

## Results and discussion

### Basic concept of CDMD

*Figure 1* summarizes the early approach proposed by *Orzechowski and Tama (2008)* for refinement of crystal structures into low-resolution electron densities. First, a simulated map $\rho_{\mathrm{sim}}(\mathbf{r})$ at a given resolution is calculated from the atomistic model and then compared with the experimental density $\rho_{\mathrm{exp}}(\mathbf{r})$ in real space. The agreement between $\rho_{\mathrm{sim}}(\mathbf{r})$ and $\rho_{\mathrm{exp}}(\mathbf{r})$ is measured by the correlation coefficient, $c.c.$, which is included into a global biasing potential $V_{\mathrm{fit}}$. This potential is added to the force field $V_{\mathrm{ff}}$ for subsequent refinement MD simulations. In this form, $V_{\mathrm{fit}}$ takes the full electron density information into account, including amplitudes and phases as well as negative densities which, by convention, reflect hydrophobic areas in the map.

To overcome sampling issues in rugged density regions, which are particularly severe given high resolutions and fine grids offered by cryo-EM today, we have extended the method to additionally allow for adaptive resolution and simulated annealing (*Figure 2*; see also *Figure 2—figure supplement 1* and Materials and methods for a detailed description of the designed protocol). The complete framework has been embedded into the GPU-accelerated GROMACS MD suite (*Abraham et al., 2015*). The refinement process is controlled by only three parameters which are varied during the MD run. First, to avoid local minima, the resolution of $\rho_{\mathrm{sim}}(\mathbf{r})$ is gradually increased from very low to the maximum available from the cryo-EM data. This increase allows the structure to adapt globally and only then locally to the experimental density during refinement. In contrast to previous approaches (*Singharoy et al., 2016*; *Wang et al., 2018*), only the simulated density is continuously resampled so that the full experimental density is used throughout the refinement. Although the correlation coefficient (*Figure 1*) is invariant with respect to whether $\rho_{\mathrm{exp}}(\mathbf{r})$ or $\rho_{\mathrm{sim}}(\mathbf{r})$ is blurred, changing the resolution of $\rho_{\mathrm{sim}}(\mathbf{r})$ only is computationally much cheaper for the biasing

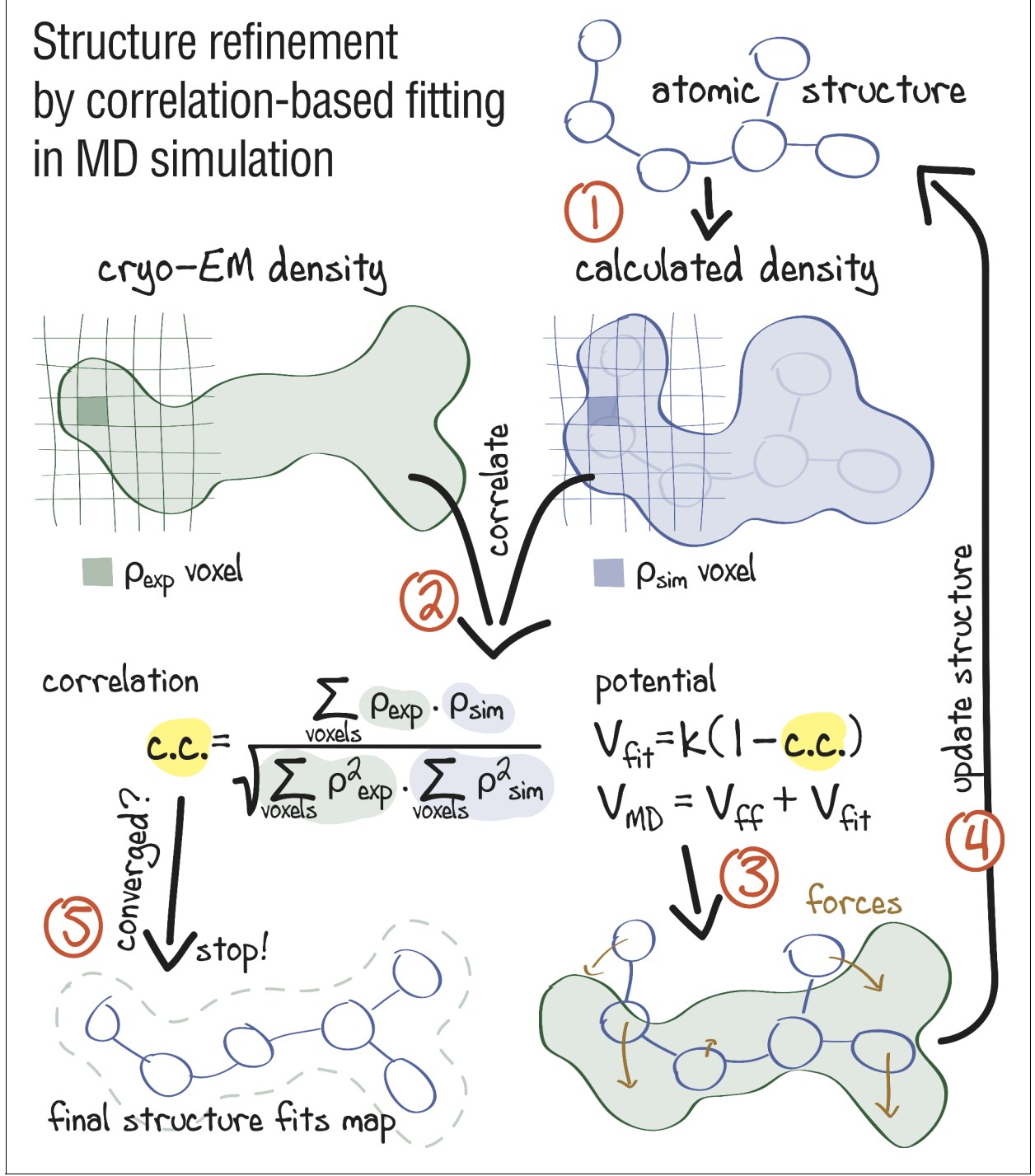

**Figure 1.** Approach: using correlation-based refinement in MD simulation to steer the atomic positions of a macromolecule such that they optimally fit a cryo-EM map. The molecule is subjected to a global biasing potential $V_{fit}$ in addition to the MD force field $V_{ff}$. The forces resulting from $V_{fit}$ act on every atom to enhance the real-space correlation coefficient $c.c.$ between the cryo-EM density (green) and the density calculated from the current atomic positions (blue). The first step (*1*) is to generate a simulated density by convoluting the atomic positions with a three-dimensional Gaussian function of width $\sigma$ (*Orzechowski and Tama, 2008*). The two maps are correlated (*2*), and the biasing forces are calculated. These forces are then added to the standard MD force field (*3*), and new atomic positions are evaluated (*4*). Steps (*1–4*) are repeated, yielding a structure that correlates better with the cryo-EM map than the starting structure.

DOI: https://doi.org/10.7554/eLife.43542.002

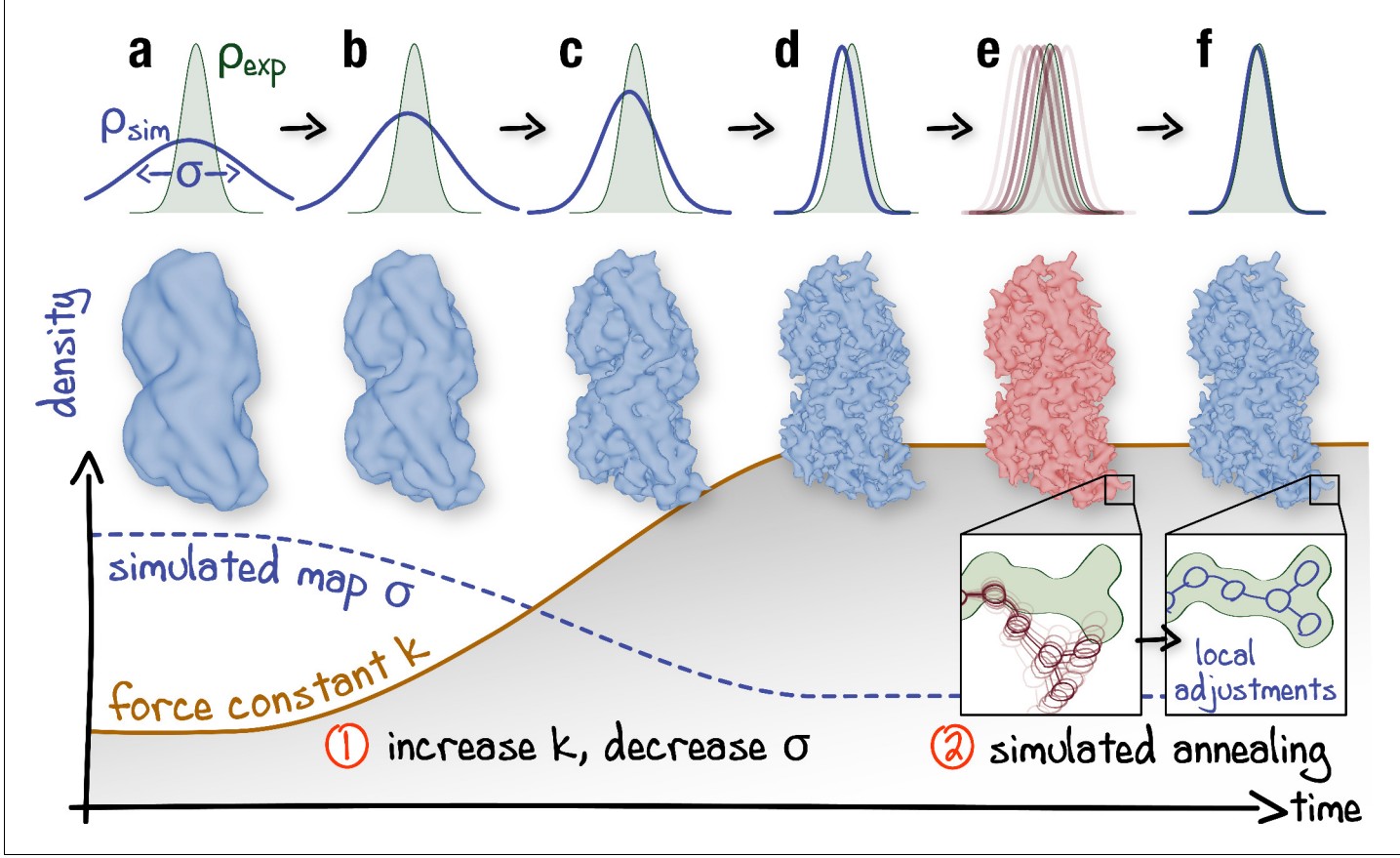

**Figure 2.** Schematic representation of the proposed continuous refinement protocol: (*1*) a low temperature optimization phase, where $V_{fit}$ is monotonously increased by increasing the force constant $k$ (columns **a–d**), followed by (*2*) simulated annealing (columns **e, f**). The local effect of the protocol is exemplified in the upper row for a one-dimensional single-atom case. Simulated densities shown in the middle row were generated using the atomic structure of a tubulin dimer (PDB ID: 3JAT; *Zhang and Nogales, 2015*).

DOI: https://doi.org/10.7554/eLife.43542.003

The following figure supplement is available for figure 2:

**Figure supplement 1.** Detailed scheme of the proposed continuous refinement protocol subdivided in five stages.

DOI: https://doi.org/10.7554/eLife.43542.004

potential of choice, which facilitates automation. Second, the relative weight of $V_{fit}$ over $V_{ff}$ is gradually increased via a force constant $k$. It has been shown previously that gradually increasing $k$ improves the refinement outcome for low-resolution maps (*Miyashita et al., 2017*). For high-resolution maps, higher force constants at late refinement stages are required to ensure an accurate density fit at the side chain level which could otherwise be distorted by thermal fluctuations. Third, high-temperature simulated annealing (*Brünger et al., 1987*; *Brunger and Adams, 2002*) is used to enhance local map-model agreement (e.g. side chain rotamers), while keeping well-refined parts of the model unchanged.

## Refinement

In this section, we describe the results for eight test cases at various resolutions (2.6–7.1 Å) covering a wide range of initial structural errors and deviations from the target state. The refined models are validated by comparing them with previously published structures. We were additionally encouraged by the reviewers to use the following alternative approaches for the high- and medium-resolution cases to compare our results to: Phenix real space refinement (*Adams et al., 2010*; *Afonine et al., 2018*), Rosetta (*DiMaio et al., 2015*; *Wang et al., 2016a*) and Refmac (*Brown et al., 2015*; *Kovalevskiy et al., 2018*). All methods are applied straightforwardly to the same far-away starting models (without manual corrections) and using protocols recommended by the developing research

groups and/or described in recent literature. While we are aware that most of these methods have *not* been originally designed or further optimized to refine distant starting models and/or DNA/ RNA, such a comparison further supports the results of this work. In the last example, we specifically compare our model with that generated by MDFF, a method originally designed for flexible fitting of distant high-resolution structures into low-resolution densities (*Trabuco et al., 2008*; *Trabuco et al., 2009*). Starting model preparation, refinement simulation setups, and validation protocols are described in Materials and methods.

Throughout the rest of the work, the following important notions will be used. (1) The *deposited reference* (or simply reference) is the atomistic model that has been generated using and deposited together with a particular cryo-EM data set in the Electron Microscopy Data Bank (EMDB). Importantly, for all cases, we use starting structures that are much farther away from the target state defined by the map than those originally used to produce the deposited references. (2) The *control structure(s)* (or simply control(s)) is a higher quality atomistic model of the same biomolecule captured in the same conformational state as the reference that has been derived from a higher resolution data set (usually X-ray). (3) The *true structure* is a hypothetical and practically unreachable structure that one would obtain from the 'perfect' data set. We generally assume that a control structure is closer to the true structure than ones derived from lower resolution cryo-EM data sets. (4) Any refinement with many parameters (here, atomic coordinates) may be prone to *overfitting*, that is refinement of the atomic coordinates into noise or map regions already occupied by the refined structure, if the strength of the biasing term is not selected appropriately. It is therefore important to refine the model against only a subset of data (*training map*) and to cross-validate it with another independent subset (*validation map*), making sure that the refinement does not lead to an overinterpretation of the training map at the cost of losing agreement with the validation map or stereochemical quality (see also *Figure 2—figure supplement 1*).

## Aldolase: good starting model against a high-resolution map

We first evaluated how well CDMD refines good starting models into high-resolution maps. Specifically, we asked if, in the absence of any other source of atomistic knowledge than the MD force field, the chosen biasing potential (*Figure 1*) per se introduces steric clashes and disrupts backbone and side chain geometries. The absence of such artifacts is an important requirement for the CDMD application at medium to low resolutions for which side chains and the secondary structure may not be sufficiently described by the density alone. To this end, our method was applied to fit the X-ray structure of a rabbit muscle aldolase (PDB ID: 6ALD; *Choi et al., 1999*) to the recently deposited data set (EMD ID: 8743, PDB ID: 5VY5; *Herzik et al., 2017*). The starting structure was subjected to MD simulation at $T$ = 300 K to increase the deviation from the target state (RMSD $\approx$ 6.6 Å; see *Figure 3a*). Higher resolution X-ray structures were used to validate the results (PDB IDs: 3BV4 at 1.7 Å (*Sherawat et al., 2008*); 1ZAH at 1.8 Å (*St-Jean et al., 2005*); 1ADO at 1.9 Å (*Blom and Sygusch, 1997*)).

Following the standard approach to avoid overfitting (*Amunts et al., 2014*; *Brown et al., 2015*), we first refined the starting model against a training map reconstructed from half of the raw cryo-EM images (half-map), while simultaneously cross-validating against a validation map (the other half-map) (*Figure 3—figure supplement 1a,b*). The target force constant was set to 5 × $10^5$ kJ mol$^{-1}$ which was roughly 10% of the total system energy and was a good initial guess. Both real-space correlation ($c.c.$) with the training map and root mean square deviation (RMSD) from the reference model (RMSD$_{ref}$) were monitored and showed a continuous improvement throughout the refinement (*Figure 3—figure supplement 1a*, bottom). Already during the half-map refinement the fitted model converged to a structure with RMSD$_{ref}$ $\approx$ 2 Å. To assess whether the refined model is overfitted, we monitored the Fourier Shell Correlation (FSC; *Harauz and van Heel, 1986*) between the model and the training map (FSC$_{train}$, shown in *Figure 3—figure supplement 1b*, top). The FSC between the model and the validation map (FSC$_{val}$) was calculated simultaneously (*Figure 3—figure supplement 1b*, bottom). No large differences were observed between FSC$_{train}$ and FSC$_{val}$ at any point during the half-map refinement, confirming the absence of overfitting. Driving the force constant beyond 5 × $10^5$ kJ mol$^{-1}$ led to gradually increasing deviations between FSC$_{train}$ and FSC$_{val}$ until the refinement simulation became unstable due to the biasing potential being too strong (15– 20% of the total system energy). Finally, we performed additional refinement including simulated

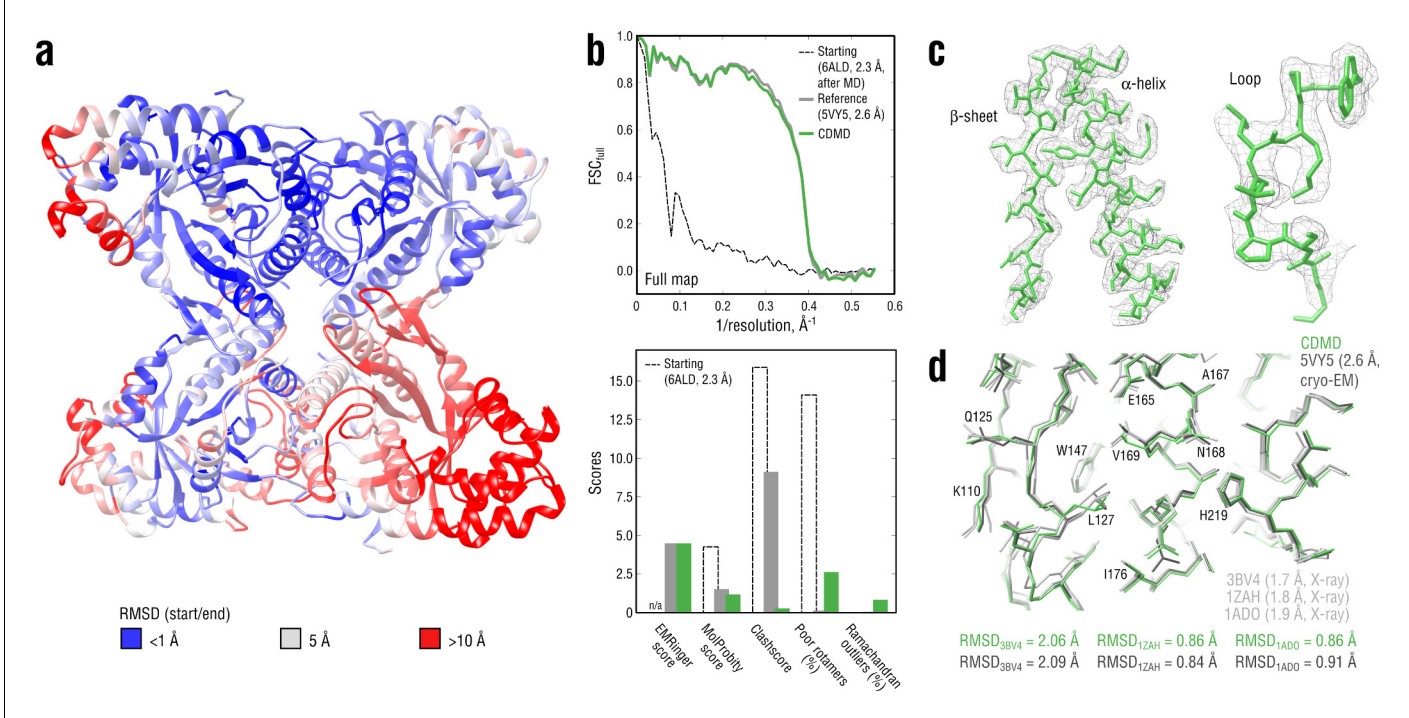

**Figure 3.** Refining a distant starting model into a high-resolution map: rabbit muscle aldolase at 2.6 Å. (**a**) RMSD (C$_\alpha$ atoms) between the starting and the reference model (5VY5) showing the extent of rearrangements during refinement. (**b**) Reciprocal-space agreement of the starting (black dashed), the reference (gray) and our refined model (green) with the full map (top) and stereochemical quality for the three models assessed by EMRinger and MolProbity (bottom). (**c**) Representative secondary structure elements showing local agreement of our model with the full map. (**d**) Representative region of the protein interior (chain A) showing the closeness of our model and the reference to higher resolution control X-ray structures in terms of RMSD. Some residues are explicitly labeled.

DOI: https://doi.org/10.7554/eLife.43542.005

The following figure supplements are available for figure 3:

**Figure supplement 1.** Extension of *Figure 3* showing the time evolution of various characteristics during refinement.

DOI: https://doi.org/10.7554/eLife.43542.006

**Figure supplement 2.** Extension of *Figure 3* showing the comparison of the radii of convergence across different refinement methods for the aldolase system and using the same distant starting structure.

DOI: https://doi.org/10.7554/eLife.43542.007

annealing against the full map to account for features not present in the training map, which brought RMSD$_{ref}$ further down to ~1.8 Å. The force constant was no longer increased.

*Figure 3b* (top) compares the starting, the deposited and our model in terms of FSC with respect to the full map (FSC$_{full}$). Stereochemical quality statistics for the three models are summarized in *Figure 3b* (bottom). Examples of secondary structure elements in the respective density parts are shown in *Figure 3c*. CDMD refinement converged quickly to a state being very close to the deposited reference and showed a significant improvement of the refined model both in geometry and reciprocal-space correlation relative to the starting structure (*Figure 3b* and *Figure 3—figure supplement 2*). Only Rosetta was able to converge as close to the reference as our method, yielding a structure with good geometry, while neither Phenix nor Refmac was. Refinement with Phenix produced reasonable geometry but showed poor convergence of solvent-exposed flexible loops, which contributed to the larger RMSD$_{ref}$. Refinement with Refmac did not place those protein regions into the density that showed the largest RMSD$_{ref}$ prior to refinement (see *Figure 3a*) and produced rather poor geometry. Stereochemical quality statistics for the models shown in *Figure 3—figure supplement 2* are summarized in *Table 1*.

When comparing our model with the deposited reference, we noticed that, despite both showing similar reciprocal correlation with the full map and having similar EMRinger scores, the latter contained almost an order of magnitude more steric clashes, whereas the rotamer and Ramachandran

**Table 1.** Refinement statistics for the aldolase system.

|  | CDMD | Phenix | Rosetta | Refmac |
|---|---|---|---|---|
| FSCavg (full map) | 0.774 | 0.704 | 0.790 | 0.778 |
| EMRinger | 4.48 | 2.64 | 4.90 | 1.31 |
| Bond lengths (Å) | 0.022 | 0.006 | 0.022 | 0.012 |
| Bond angles (°) | 2.22 | 1.30 | 1.76 | 2.90 |
| MolProbity | 1.15 | 1.49 | 0.61 | 3.57 |
| All-atom clashscore | 0.24 | 3.82 | 0.28 | 31.6 |
| Ramachandran statistics: | | | | |
| Favored (%) | 96.41 | 100.0 | 98.17 | 93.33 |
| Allowed (%) | 2.79 | 0.0 | 1.76 | 5.06 |
| Outliers (%) | 0.81 | 0.0 | 0.07 | 1.61 |
| Poor rotamers (%) | 2.61 | 2.61 | 0.09 | 32.6 |
| CaBLAM flagged (%) | 9.0 | 10.1 | 8.32 | 13.3 |

DOI: https://doi.org/10.7554/eLife.43542.029

statistics were poorer for the former. We assume that this relationship between the number of steric clashes and the rotamer/backbone quality may be explained by the way CDMD and those methods relying on knowledge-based structural restraints (e.g. Rosetta or Phenix) optimize the fit to density. We postpone a more detailed discussion on this observation to the end of the Results section. To determine which of the two models was in fact closer to the 'true structure', we compared both with the higher-quality control structures (*Figure 3d*). Neither model showed a considerably smaller RMSD value to any of the control structures, suggesting that both models might be equally good approximations of the 'true structure'. Overall, the aldolase results suggest that CDMD is capable of correcting structural errors and converges quickly to the target state while maintaining good geometry at all refinement stages.

## Tubulin: accuracy and convergence at medium resolutions

We next applied CDMD to a medium-resolution map (4.1 Å) of a GDP-tubulin dimer in the straight microtubule-like conformation derived from an asymmetric helical reconstruction of a complete, kinesin-decorated microtubule lattice (provided by courtesy of *Zhang and Nogales, 2015*). No reference structure was available for this map. However, two high-resolution GDP-tubulin structures (PDB ID: 3JAS at 3.3 Å and PDB ID: 6DPV at 3.5 Å) in the same conformational state were available (*Zhang et al., 2015*; *Zhang et al., 2018*). We used these structures as controls to confirm the method's accuracy and convergence for medium-resolution maps. A curved structure of GDP-tubulin (PDB ID: 4ZOL; *Wang et al., 2016b*) after ~3 μs of MD simulation in explicit solvent (*Igaev and Grubmüller, 2018*) was used as the starting model (RMSD$_{6DPV}$ ≈ 3.7 Å; C$\alpha$ atom deviations from the target straight state are shown in *Figure 4a*).

*Figure 4b,c,d* compares our model with the two high-resolution control structures as well as with the starting model. Despite the higher accuracy of the control structures, our model reached the same reciprocal-space correlation with the full map across the whole range of spatial frequencies (*Figure 4b*, top). This is particularly remarkable because the high-resolution features were not as pronounced in the 4.1 Å map as they were in the 3.3 Å and 3.5 Å maps used for generating the control structures. The good correlation did not result from overfitting or geometry violations as confirmed by the cross-validation procedure (as in *Figure 3—figure supplement 1b*, data not shown) and stereochemical quality assessment (*Figure 4b*, bottom). Our model was very similar in stereochemical quality to the control structures except for the ~10-fold smaller number of steric clashes and a higher fraction of Ramachandran outliers. The higher EMRinger score for our model indicated improved local side chain correlations and backbone placement.

As in the aldolase case, CDMD refinement quickly converged to the straight state that was very similar in RMSD to the control structures (*Figure 4—figure supplement 1*). Among all the methods tested, only Phenix produced a structure with comparable geometry and map agreement as our

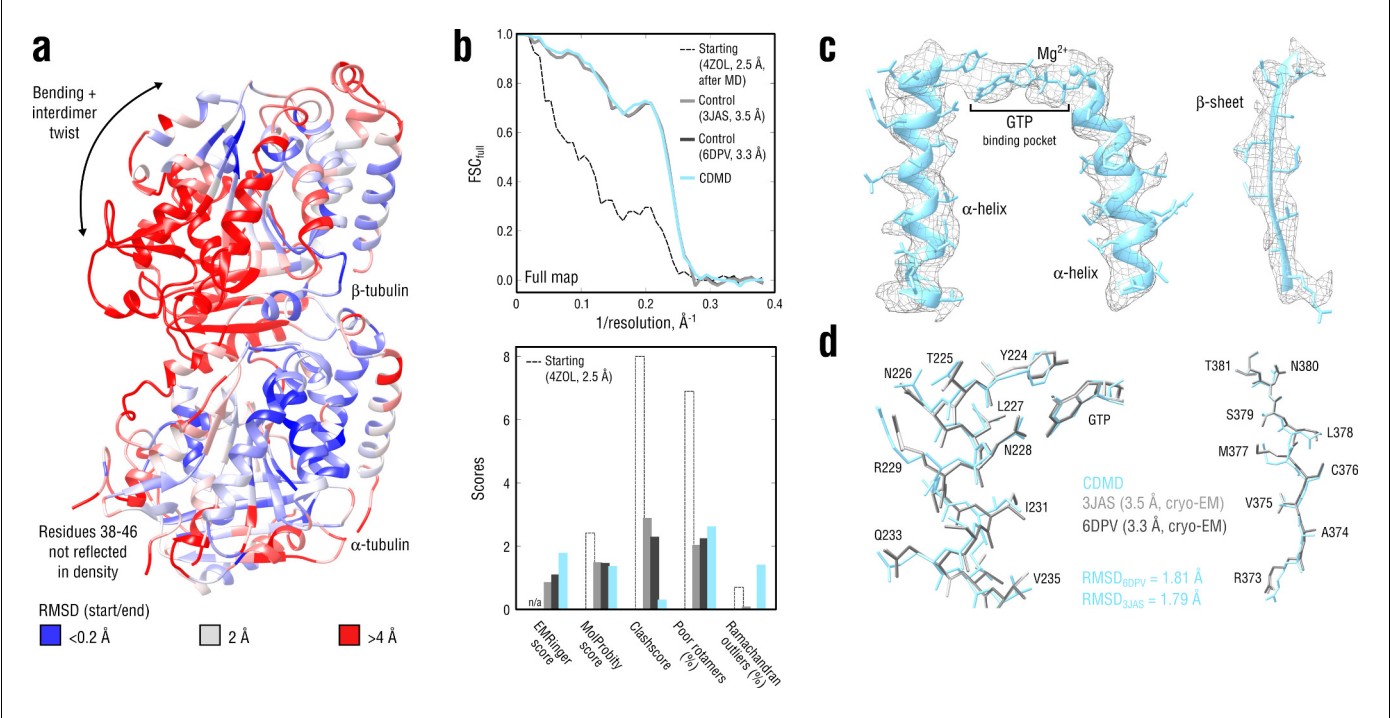

**Figure 4.** Refinement of a curved tubulin dimer into a map of the straight, microtubule-like state. (a) RMSD ($C_\alpha$ atoms) between the starting and the control model (6DPV) showing the extent of rearrangements between the solution (curved) and the microtubule-like (straight) tubulin conformation. The α-subunits of both models were aligned for the RMSD calculation. (b) Reciprocal-space agreement of the starting (black dashed), the control (gray and dark gray) and our model (cyan) with the full map (top) and stereochemical quality for the four models assessed by EMRinger and MolProbity (bottom). (c) Representative secondary structure elements showing local agreement of our model with the full map. (d) Same as in c but showing the closeness of our model to higher resolution control structures in terms of RMSD. Some residues are explicitly labeled.

DOI: https://doi.org/10.7554/eLife.43542.008

The following figure supplement is available for figure 4:

**Figure supplement 1.** Extension of *Figure 4* showing the comparison of the radii of convergence across different refinement methods for the tubulin system and using the same distant starting structure.

DOI: https://doi.org/10.7554/eLife.43542.009

model. Rosetta and Refmac showed poorer convergence in more flexible loop regions, with the Rosetta model having a much higher number of steric clashes. Stereochemical quality statistics for the models shown in *Figure 4—figure supplement 1* are summarized in *Table 2*. Altogether, these results demonstrate the accuracy and good convergence of CDMD at medium resolutions, namely its ability to provide plausible local geometries in cases where this information is not fully present in the map and the refinement has to partially rely on the force field.

## TRPV1: highly heterogeneous local resolution

Our next test system was the vanilloid receptor 1 (TRPV1) resolved at 3.2 Å (EMD ID: 5778, PDB ID: 3J5P; *Liao et al., 2013*) for which the local resolution ranged from 2.5 to 7 Å, such that no standard non-MD refinement algorithm could be applied to fit a distant model uniformly well to all map regions. The large difference in the local resolution is due to a rigid transmembrane (TM) domain and flexible, cytosolic ankyrin repeat domains (ARDs) (*Liao et al., 2013*; *Gao et al., 2016*). To further challenge our method, we generated a distant starting model with poor geometry by subjecting the reference structure (3J5P) to MD simulation in explicit solvent at $T = 300$ K. The overall $RMSD_{ref}$ for the selected starting model was ~5 Å, with local deviations reaching up to ~20 Å in the ARD region and the upper part of the TM domain (*Figure 5a*). No higher resolution TRPV1 structures were available for additional model validation. It should also be noted that residues 111–198 of the ARDs were only poorly reflected in the density. We nevertheless included these parts in the refinement

**Table 2.** Refinement statistics for the tubulin system.

| | CDMD | Phenix | Rosetta | Refmac |
|---|---|---|---|---|
| FSCavg (full map) | 0.756 | 0.784 | 0.703 | 0.743 |
| EMRinger | 1.79 | 1.07 | 0.92 | 0.72 |
| Bond lengths (Å) | 0.022 | 0.010 | 0.021 | 0.014 |
| Bond angles (°) | 2.30 | 1.57 | 2.79 | 2.15 |
| MolProbity | 1.42 | 1.58 | 2.30 | 2.58 |
| All-atom clashscore | 0.46 | 11.7 | 24.9 | 13.8 |
| Ramachandran statistics: | | | | |
| Favored (%) | 93.28 | 99.41 | 93.75 | 93.16 |
| Allowed (%) | 5.31 | 0.59 | 4.72 | 6.01 |
| Outliers (%) | 1.42 | 0.0 | 1.53 | 0.83 |
| Poor rotamers (%) | 2.63 | 0.73 | 0.14 | 4.38 |
| CaBLAM flagged (%) | 14.1 | 16.3 | 16.8 | 9.5 |

DOI: https://doi.org/10.7554/eLife.43542.030

procedure because the correlation-driven potential (*Figure 1*) does not depend on the local density amplitude and would still refine low-resolution features (overall shape, orientation, etc.).

*Figure 5b* compares our model with the deposited reference structure as well as with the starting model. Unlike in the above cases, our model correlated better with the full map at spatial frequencies below 0.1 Å$^{-1}$ (better rigid-body positioning of the domains) but worse in the 0.2–0.3 Å$^{-1}$ range (*Figure 5b*, top). We hence asked whether this difference in the higher frequency range might have resulted from the deposited structure (PDB ID: 3J5P) having conflicting geometries or from our model being underfitted due to a low force constant (here, $k = 4 \times 10^5$ kJ mol$^{-1}$). Neither the standard cross-validation procedure (as in *Figure 3—figure supplement 1*, data not shown) nor subsequent re-refinement of the starting model using higher force constants (4.5 and 5 × 10$^5$ kJ mol$^{-1}$) revealed any over- or underfitting, which suggests that the seemingly better correlation of the deposited model at higher spatial frequencies is likely a result of overfitting. This was further supported by the model quality assessment (*Figure 5b*, bottom), where our model was systematically better in all quality statistics except for a moderate fraction of Ramachandran outliers.

To determine the radius of convergence for the methods under study, we performed comparative modeling using the same distant starting structure as above (*Figure 5—figure supplement 1*). CDMD refinement converged to a structure with good geometry and RMSD$_{ref}$ ≈ 2.5 Å, while the other methods did not. The convergence was good in the well-resolved TM region across all methods, but only Refmac was able to place the flexible ARD regions back into the density (RMSD$_{ref}$ ≈ 2.8 Å), however, at the price of severe geometry violations. Phenix and Rosetta yielded accurate models but were unable to refine solvent-exposed flexible loops in the ARD region. Stereochemical quality statistics for the models shown in *Figure 5—figure supplement 1* are summarized in *Table 3*.

Finally, to assess to what extent the CDMD refinement depends on the local map resolution, we performed two additional refinement runs using different but similarly distant starting structures (RMSD$_{ref}$ ≈ 5 Å in both cases). We found that all three refinements produce very similar models but the structural variability was larger in the ARDs, whereas the refinements converged to almost the same structure in the TM domain, consistent with the local resolution in these regions (*Figure 5—figure supplement 2*). In summary, the TRPV1 refinements demonstrate that CDMD has a large and robust radius of convergence and is able to significantly improve poor starting models irrespective of local resolution and within a single refinement run.

## TRPV1: comparison with Rosetta and ReMDFF

The TRPV1 system has been previously used to benchmark the Rosetta (*Wang et al., 2016a*) and ReMDFF (*Singharoy et al., 2016*; *Wang et al., 2018*) algorithms, which offers an additional comparison to these methods. However, the Rosetta model (PDB ID: 3J9J; *Barad et al., 2015*) contained

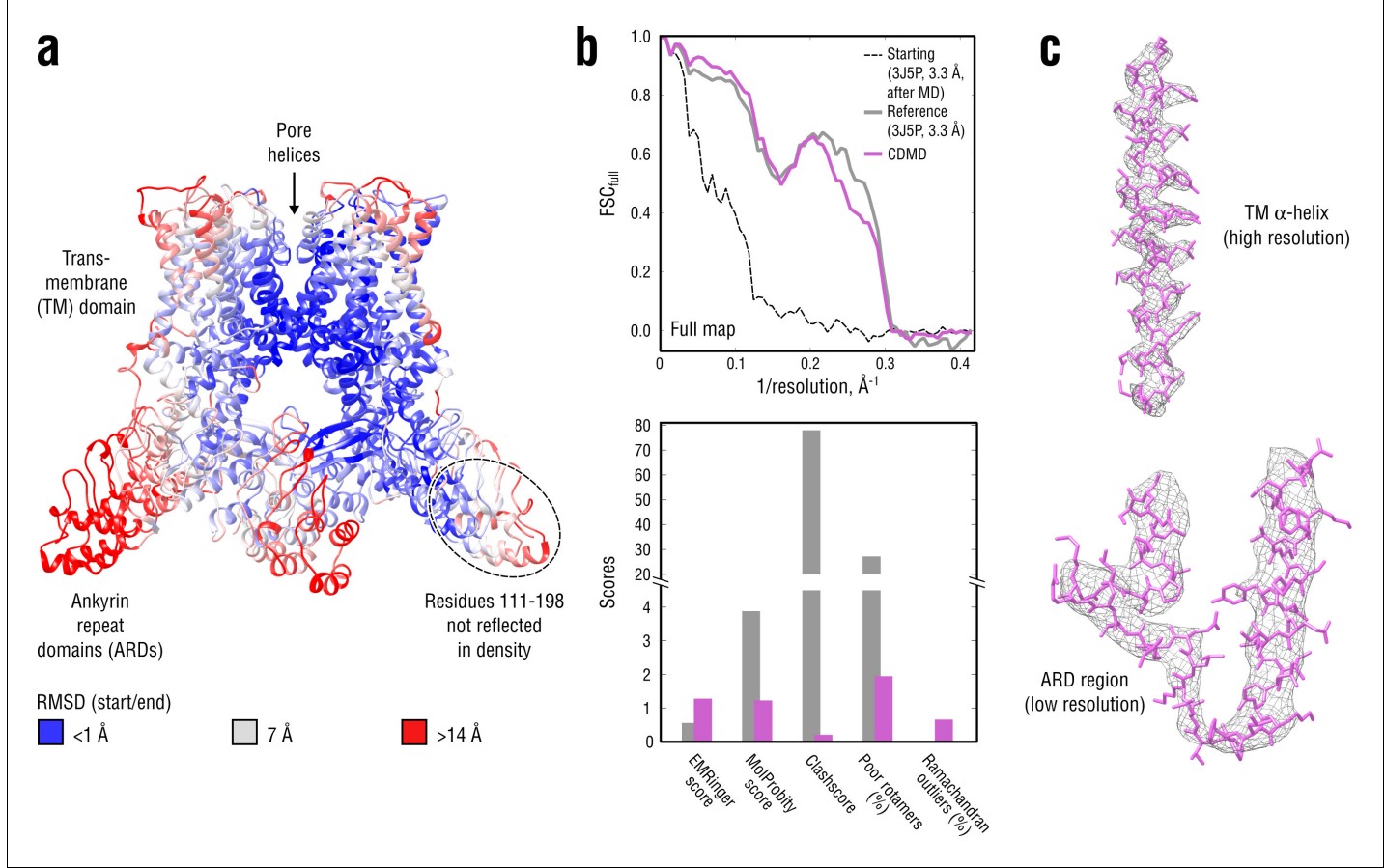

**Figure 5.** Refinement of a poor structure of TRPV1 in a distant conformation into a map with highly heterogeneous local resolution. (a) RMSD ($C_\alpha$ atoms) between the starting and the reference model (3J5P) showing the extent of rearrangements the TRPV1 structure undergoes during refinement. (b) Reciprocal-space agreement of the starting (black dashed), the reference (gray) and our model (purple) with the full map (top) and stereochemical quality of the models assessed by EMRinger and MolProbity (bottom). Note that the starting model was derived from the deposited structure by subjecting the latter to MD at $T$ = 300 K. (c) Representative secondary structure elements showing local agreement between of our model with the full map.

DOI: https://doi.org/10.7554/eLife.43542.010

The following figure supplements are available for figure 5:

**Figure supplement 1.** Extension of *Figure 5* showing the comparison of the radii of convergence across different refinement methods for the TRPV1 system and using the same distant starting structure.

DOI: https://doi.org/10.7554/eLife.43542.011

**Figure supplement 2.** Convergence of the TRPV1 refinement both in the higher resolution TM region and in the lower resolution ARD region assessed by means of three independent refinement runs using different but similarly distant starting structures.

DOI: https://doi.org/10.7554/eLife.43542.012

only the TM region, whereas only one TRPV1 monomer was refined with ReMDFF (courtesy by *Wang et al., 2018*). For a direct comparison, we cut off the flexible ARDs in the starting TRPV1 model (*Figure 5a*) to match the sequence of the deposited Rosetta model and re-refined it using a higher force constant ($k$ = 5 × 10$^5$ kJ mol$^{-1}$), as more structural details and less structural variability were expected for the high-resolution TM region (*Figure 6a,b*). For the ReMDFF comparison (*Figure 6c,d*), no additional refinement was performed, and the monomer showing the best map-model agreement and model geometry from the three independently refined TRPV1 structures was chosen for further analysis (*Figure 5—figure supplement 2*).

As shown in *Figure 6b*, our model for the TM region compared well with that generated by Rosetta both in terms of reciprocal-space correlation and model quality. In the ReMDFF comparison, our model outperformed the ReMDFF model both in reciprocal-space correlation and model

**Table 3.** Refinement statistics for the TRPV1 system.

|  | CDMD | Phenix | Rosetta | Refmac | CDMD (TM domain) |
|---|---|---|---|---|---|
| FSCavg (full map) | 0.632 | 0.639 | 0.626 | 0.530 | 0.684 |
| EMRinger | 1.28 | 1.08 | 0.98 | 0.348 | 2.12 |
| Bond lengths (Å) | 0.022 | 0.008 | 0.021 | 0.012 | 0.024 |
| Bond angles (°) | 2.14 | 1.50 | 2.18 | 2.81 | 2.25 |
| MolProbity | 1.23 | 1.90 | 1.39 | 3.55 | 1.50 |
| All-atom clashscore | 0.21 | 8.40 | 2.03 | 29.0 | 0.29 |
| Ramachandran statistics: |  |  |  |  |  |
| Favored (%) | 93.37 | 99.24 | 93.75 | 91.29 | 90.89 |
| Allowed (%) | 5.97 | 0.76 | 5.26 | 7.01 | 7.59 |
| Outliers (%) | 0.66 | 0.0 | 0.99 | 1.70 | 1.52 |
| Poor rotamers (%) | 1.95 | 3.90 | 0.11 | 26.9 | 3.01 |
| CaBLAM flagged (%) | 15.9 | 23.2 | 20.5 | 17.9 | 14.8 |

DOI: https://doi.org/10.7554/eLife.43542.031

geometry (*Figure 6d*). This was mainly due to better map-model agreement of our model in the low-resolution ARD region (*Figure 6c*, bottom insert), which was confirmed by the higher reciprocal correlation values at low spatial frequencies (*Figure 6d*, top). We also observed much poorer rotamer and Ramachandran distributions for the ReMDFF model (*Figure 6d*, bottom) despite the fact that, like our method, ReMDFF employs an accurate atomistic force field.

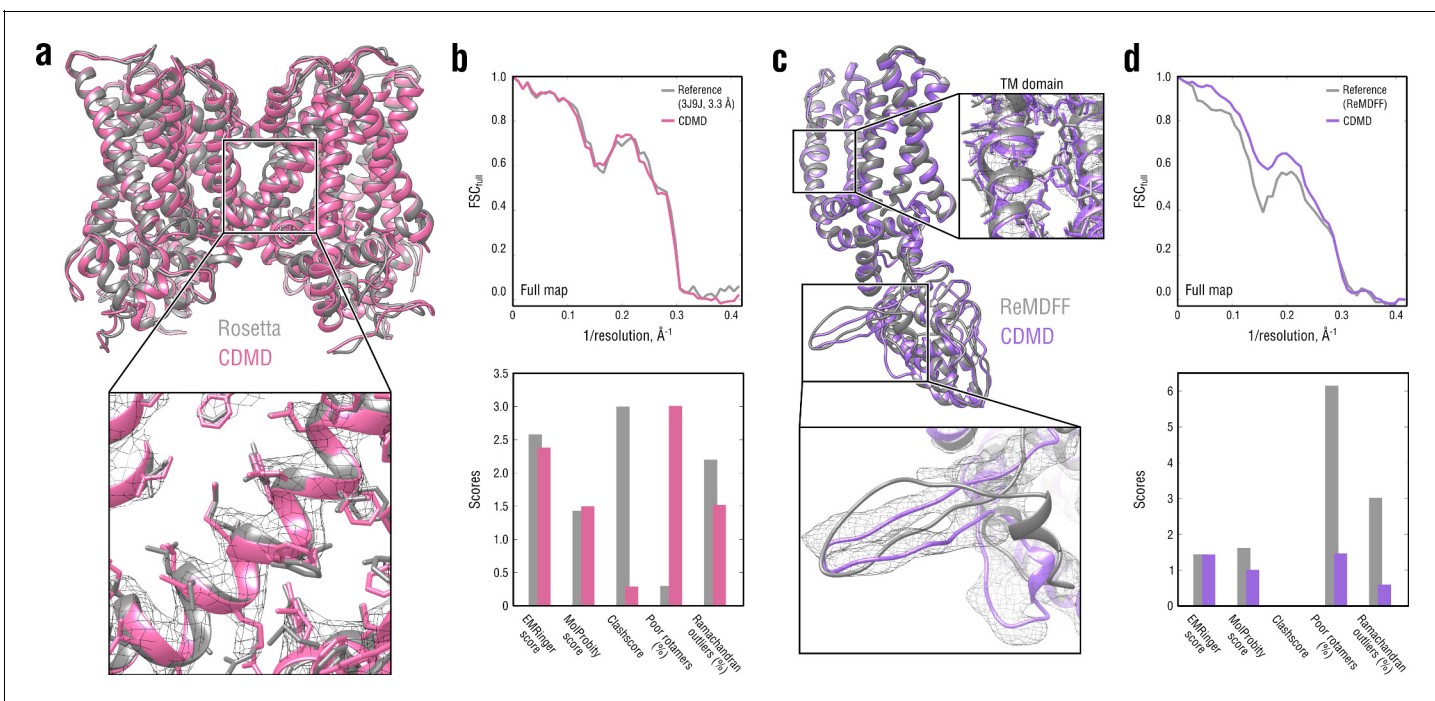

**Figure 6.** Comparison of our TRPV1 model with those previously refined using Rosetta (**a, b**) and ReMDFF (**c, d**). Overlays of our model (pink and violet ribbon) with the Rosetta (left, gray ribbon) and ReMDFF (right, gray ribbon) models are shown in (**a**) and (**c**), respectively. Reciprocal-space agreement with the full map (top) and stereochemical quality for the four models assessed by EMRinger and MolProbity (bottom) are shown in (**b**) for the Rosetta model and in (**d**) for the ReMDFF model.

DOI: https://doi.org/10.7554/eLife.43542.013

## N-ethylmaleimide sensitive factor (NSF): comparison with Phenix

Further to the previous section, we compared how CDMD performs on a system that has been recently refined using the newest Phenix protocol (*Afonine et al., 2018*) with optimized refinement parameters (*White et al., 2018*). To this end, the structure of an ATP-bound NSF (PDB ID: 3J94; *Zhao et al., 2015*) was refined into the ~3.9 Å map of a NSF complex (EMD ID: 9102, class 1; *White et al., 2018*). To enable a direct comparison, the starting structure was completed to match the deposited Phenix model (see Materials and methods). The starting structure was also subjected to MD simulation at $T$ = 300 K to increase the deviation from the target state (*Figure 7a*). Higher-resolution X-ray structures were used to validate the results (PDB IDs: 1NSF at 1.9 Å (*Yu et al., 1998*); 1D2N at 1.75 Å (*Lenzen et al., 1998*)). No half-maps were available for this test case, which precluded the half-map-based cross-validation and, hence, determining the optimal force constant. We therefore performed seven independent refinement runs starting from the same distant structure (*Figure 7a*) and using a wide range of force constants (2–8 $\times$ $10^5$ kJ mol$^{-1}$). The model with the best balance between map-model agreement and geometry was then used for further analysis (see Materials and methods for details on refinements against full maps only).

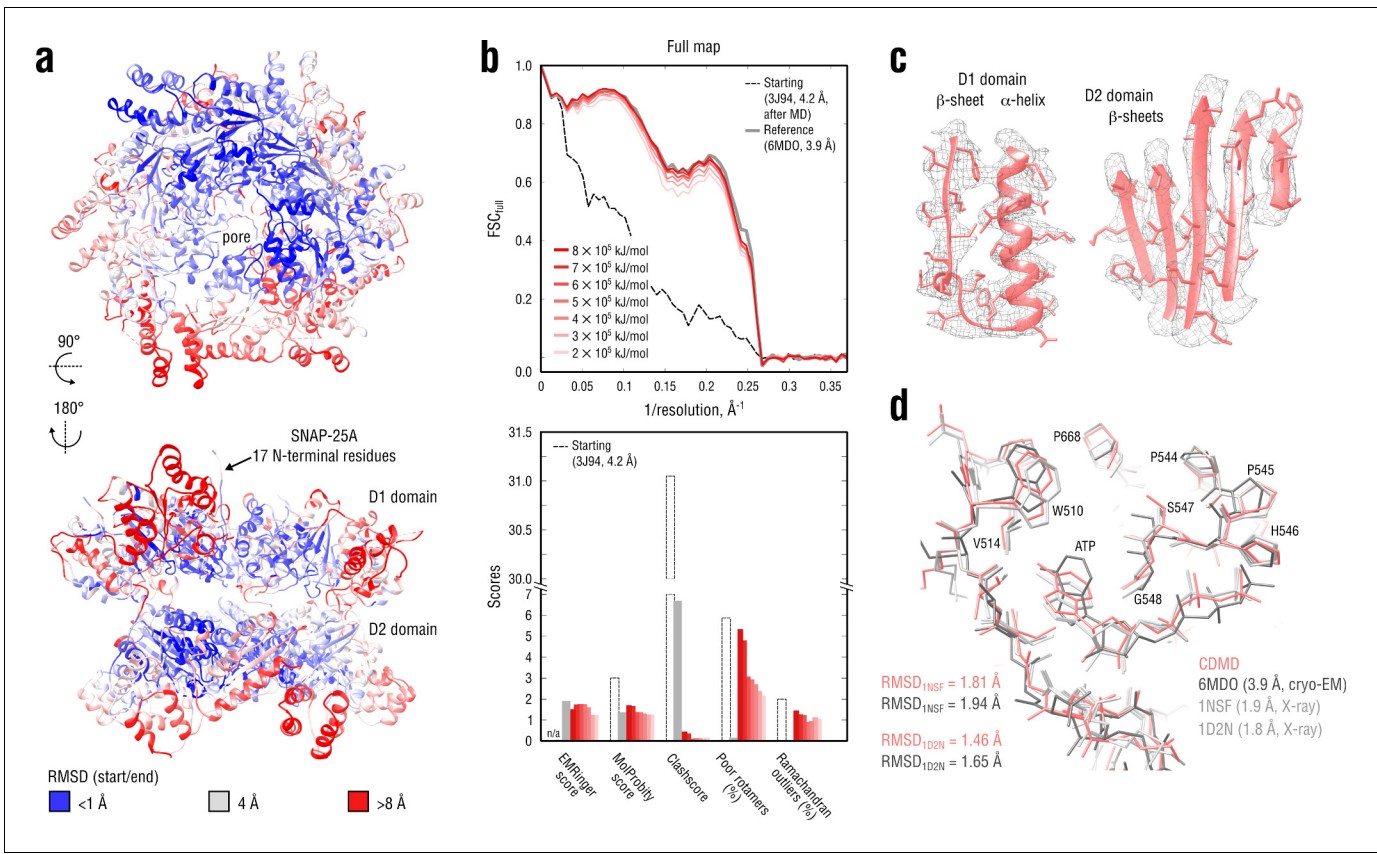

**Figure 7.** Refinement of a substrate-free NSF complex in a distant conformation into a medium-resolution map at 3.9 Å. (**a**) RMSD (C$_\alpha$ atoms) between the starting and the reference model (6MDO) showing the extent of rearrangements the NSF structure undergoes during refinement. (**b**) Reciprocal-space agreement with the full map for the starting (black dashed), the reference (gray) and the set of final models (red gradient) refined using a wide range of target force constants (top) and stereochemical quality assessed by EMRinger and MolProbity (bottom). (**c**) Representative secondary structure elements showing local agreement of the model refined at $k$ = 5 $\times$ $10^5$ kJ mol$^{-1}$ with the map. (**d**) ATP binding pocket of the D2 domain (chain A) showing the closeness of our model and the reference to higher-resolution control X-ray structures in terms of RMSD. Some residues are explicitly labeled.

DOI: https://doi.org/10.7554/eLife.43542.014

The following figure supplement is available for figure 7:

**Figure supplement 1.** Extension of *Figure 7* showing the comparison of the radii of convergence across different refinement methods for the NSF system and using the same distant starting structure.

DOI: https://doi.org/10.7554/eLife.43542.015

**Table 4.** Refinement statistics for the NSF system.

| | CDMD | Phenix | Rosetta | Refmac |
|---|---|---|---|---|
| FSCavg (full map) | 0.765 | 0.706 | 0.748 | 0.479 |
| EMRinger | 1.77 | 0.90 | 1.56 | 0.11 |
| Bond lengths (Å) | 0.022 | 0.007 | 0.021 | 0.020 |
| Bond angles (°) | 2.30 | 1.52 | 2.17 | 3.29 |
| MolProbity | 1.38 | 1.38 | 1.42 | 3.78 |
| All-atom clashscore | 0.13 | 6.93 | 3.26 | 47.1 |
| Ramachandran statistics: | | | | |
| Favored (%) | 92.37 | 98.86 | 95.67 | 89.53 |
| Allowed (%) | 6.71 | 1.03 | 3.73 | 7.03 |
| Outliers (%) | 0.92 | 0.11 | 0.60 | 3.44 |
| Poor rotamers (%) | 2.95 | 0.41 | 0.08 | 25.8 |
| CaBLAM flagged (%) | 13.1 | 20.5 | 14.9 | 22.1 |

DOI: https://doi.org/10.7554/eLife.43542.032

*Figure 7b* compares our refined NSF models with the deposited reference as well as with the starting model in terms of map-model agreement and geometry. *Figure 7c* additionally shows local agreement of our model with the full map. The map-model agreement improved gradually as the target force constant increased, but the final models correlated almost equally well with the full map (*Figure 7b*, top, red curves). Because the half-maps were not available, it was challenging to judge which of the obtained models suffered from under- or overfitting. We therefore assessed the stereochemistry of the models (*Figure 7b*, bottom), while simultaneously comparing them to the deposited reference structure (PDB ID: 6MDO). One would expect that, as the force constant becomes too high, the map-model agreement would increase at the expense of geometry violations (atomic clashes or rotamer and Ramachandran outliers), whereas for insufficiently strong force constants, the model stereochemistry should be less dependent on the biasing force, suggesting that the 'optimal' force constant should fall within the chosen range ($2$–$8 \times 10^5$ kJ mol$^{-1}$). Indeed, this was the case for the NSF refinement: while the models at $k = 2$–$6 \times 10^5$ kJ mol$^{-1}$ showed similar geometry, increasing the force constant beyond $6 \times 10^5$ kJ mol$^{-1}$ led to a drastic increase in the number of rotamer outliers comparable with that of the starting model, potentially indicating overfitting. We also noticed that the models at 2 and $3 \times 10^5$ kJ mol$^{-1}$ had lower EMRinger scores, suggesting that the biasing force was insufficient to improve the local side chain-map correlation and potentially indicating underfitting. This was further supported by the weaker reciprocal correlation for these two models as compared to both reference and other models at $k > 3 \times 10^5$ kJ mol$^{-1}$ (*Figure 7b*, top). We hence concluded that the optimal force constant should be in the range of $4$–$5 \times 10^5$ kJ mol$^{-1}$, which was comparable to what we used to refine the TRPV1 system (*Figure 5*) being approximately of the same size. We hence used the structure refined at $5 \times 10^5$ kJ mol$^{-1}$ for further analysis.

Comparison of our NSF model with the deposited reference revealed that it had a substantially lower clashscore (*Figure 7b*, bottom), which might be attributed to the use of an accurate force field that improved poorly defined regions. To determine whether this difference also indicated that our NSF model was closer to the 'true structure', we compared both models with the higher quality X-ray structures in terms of RMSD (*Figure 7d*). As the control structures contained only a single protomer of the D2 ring, we truncated our model and the reference to match each of the control structures prior to RMSD calculation. Our NSF model showed systematically lower RMSD values to both controls (roughly a 0.2 Å difference, which is detectable given the small size of the compared structures). Visual inspection revealed that this was mainly explained by the difference in the ATP binding pockets and that our model more closely resembled the control structures in this region. Particularly, we noticed that either the adenosine group of ATP was rotated by 90° or 180° relative to that of the controls (3 of 6 NSF protomers in 6MDO) or the phosphate tail of ATP deviated from that of the controls (all protomers in 6MDO). In contrast, our model reproduced the structure of the ATP

binding pocket equally well across the entire D2 hexamer. We speculate that a proper treatment of electrostatics is important for correct ligand placement in the absence of sufficient density information.

Finally, we assessed the convergence of the NSF refinement across the different methods (*Figure 7—figure supplement 1*). Stereochemical quality statistics for the models are summarized in *Table 4*. Unlike in the above test cases, none of the alternative refinement methods was able to approach the reference structure demonstrating only ~0.5 Å of improvement in terms of RMSD$_{ref}$ as compared to ~2 Å of improvement by CDMD. Interestingly, the convergence was better in the D2 domain for all alternative methods than in the D1 domain, where the split region between protomers A and F and the SNAP-25A 17 N-terminal residues were only poorly modelled by Phenix, Rosetta and Refmac. We speculate that, for the alternative methods, the hexagonal symmetry of the D2 domain translated into global symmetry restraints improves the refinement of this part of the NSF complex relative to the D1 domain, where the symmetry is broken by the split region. Here, we

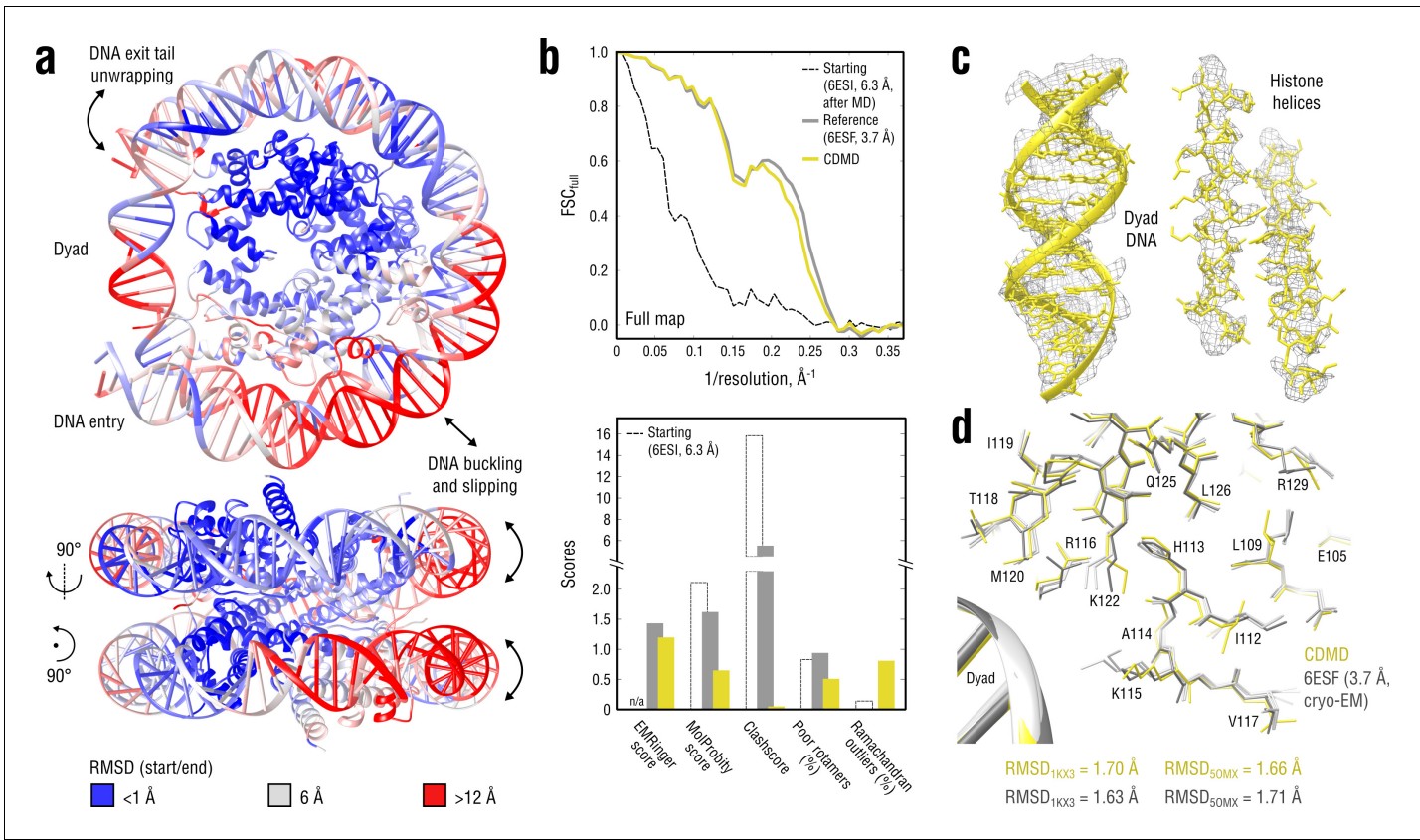

**Figure 8.** Refinement of a distant nucleosome structure into a medium-resolution map of the canonical nucleosome state. (a) RMSD (DNA and protein backbone) between the starting and the reference model (6ESF) showing the extent of rearrangements during the refinement. (b) Reciprocal-space agreement of the starting (black dashed), the reference (gray) and our model (yellow) with the full map (top) and stereochemical quality assessed by EMRinger and MolProbity (bottom). (c) Representative secondary structure elements showing local agreement of our model with the full map. (d) Representative region next to the dyad DNA showing the closeness of our model to higher-resolution control structures in terms of RMSD (only protein non-hydrogen atoms). Some residues are explicitly labeled.

DOI: https://doi.org/10.7554/eLife.43542.016

The following figure supplements are available for figure 8:

**Figure supplement 1.** Extension of *Figure 8b* showing the reciprocal-space agreement and stereochemical quality for nucleosome models independently refined using force constants ranging from 2 to 4.5 × 10$^5$ kJ mol$^{-1}$.
DOI: https://doi.org/10.7554/eLife.43542.017

**Figure supplement 2.** Extension of *Figure 8* showing the comparison of the radii of convergence across different refinement methods for the nucleosome system and using the same distant starting structure.
DOI: https://doi.org/10.7554/eLife.43542.018

again emphasize that CDMD does not need symmetry restraints (or any other geometry-based restraints) and relies solely on the interatomic interactions defined by the force field.

## Nucleosome: non-rigid-body transition in a protein-DNA complex

We next sought to test how CDMD would perform on a system (a) for which a non-trivial concerted motion is expected (i.e. which cannot be described by a set of rigid body translations or rotations), and (b) that also includes DNA whose correct stereochemistry is notoriously hard to maintain using modern force fields (*Galindo-Murillo et al., 2016*; *Ivani et al., 2016*; *Dans et al., 2017*). To this end, we refined the recently published low-resolution structure of a 'breathing' nucleosome complex (PDB ID: 6ESI; *Bilokapic et al., 2018*) into the 3.7 Å map of the canonical nucleosome state (EMD ID: 3947, PDB ID: 6ESF; *Bilokapic et al., 2018*). The major rearrangements between the two states include a contraction along the symmetry axis coupled to a concomitant expansion in the perpendicular direction and a shift by ~1 base pair in the DNA due to rearrangements in the histone-DNA interface. To further challenge our refinement method, we increased the RMSD of the starting structure from the target state by subjecting it to MD simulation in explicit solvent at $T$ = 300 K, which, in addition to the above deviations, led to overall DNA buckling and slipping off the histone octamer (RMSD ≈ 5 Å with local deviations reaching ~15 Å; see *Figure 8a*). The success of refining such a starting structure, therefore, ultimately hinges on an accurate treatment of concerted molecular motions. We used the following higher-quality X-ray structures of the canonical nucleosome state to validate the results: 1KX3 at 1.9 Å (*Davey et al., 2002*) and 5OMX at 2.3 Å (*Frouws et al., 2018*) for the histone part only (due to different DNA sequences), and 5MLU at 2.8 Å (*Makde et al., 2010*) for the DNA part.

*Figure 8b* compares our model with the deposited reference in terms of map-model agreement and geometry. *Figure 8c* additionally shows local agreement of our model with the full map. Both models correlated almost equally well with the full map (*Figure 8b*, top), with the deposited model correlating slightly better at spatial frequencies above 0.2 Å$^{-1}$, which again raised the question of underfitting as in the case of the TRPV1 refinement (see previous section). We therefore performed a series of nucleosome refinements using force constants ranging from 2 to 4.5 $\times$ 10$^5$ kJ mol$^{-1}$ ($k$ = 3 $\times$ 10$^5$ kJ mol$^{-1}$ was used in *Figure 8*), while simultaneously assessing the model geometry (*Figure 8—figure supplement 1*). The goodness-of-fit slowly improved as the force constant increased, but the refined structure never reached the same extent of reciprocal-space correlation as the deposited reference, suggesting that the deposited reference structure might be slightly overfitting the full map. This conclusion was also supported by the model quality statistics (*Figure 8b*, bottom) which showed a systematic improvement of our model relative to both starting and deposited reference structure, except for a moderate fraction of Ramachandran outliers and a slightly lower EMRinger score.

**Table 5.** Refinement statistics for the nucleosome system.

| | CDMD | Phenix | Rosetta | Refmac |
|---|---|---|---|---|
| FSCavg (full map) | 0.659 | 0.676 | 0.572 | 0.399 |
| EMRinger | 1.20 | 1.50 | 0.88 | 0.76 |
| Bond lengths (Å) | 0.022 | 0.008 | 0.019 | 0.018 |
| Bond angles (°) | 1.82 | 1.10 | 2.60 | 3.38 |
| MolProbity | 0.65 | 1.58 | 2.60 | 3.88 |
| All-atom clashscore | 0.1 | 8.84 | 91.7 | 63.5 |
| Ramachandran statistics: | | | | |
| Favored (%) | 97.31 | 99.87 | 97.04 | 88.84 |
| Allowed (%) | 1.88 | 0.13 | 1.75 | 8.60 |
| Outliers (%) | 0.81 | 0.0 | 1.21 | 2.55 |
| Poor rotamers (%) | 0.51 | 1.36 | 0.16 | 22.62 |
| CaBLAM flagged (%) | 5.6 | 8.2 | 9.4 | 15.3 |

DOI: https://doi.org/10.7554/eLife.43542.033

To validate these results, we compared our model and the deposited reference with the higher resolution control structures in terms of RMSD (*Figure 8d*, only protein non-hydrogen atoms). As in the aldolase case, neither model had systematically lower RMSD values to both control structures, suggesting that both models are good representations of the 'true structure'. The DNA part was very similar across all models: $RMSD_{control} \approx 1.4$ Å for both our model and the reference, and RMSD less than 0.9 Å between the compared models themselves (all DNA non-hydrogen atoms). It should be noted that the DNA accuracy in the deposited reference (refined with Phenix) was due to the use of base-pairing and base-stacking restraints and a much closer starting structure. On the contrary, the high level of achieved DNA quality in our model is particularly remarkable because (a) no geometry restraints were used, and (b) the CHARMM force field family used to refine the nucleosome (see Materials and methods) is rather inaccurate in reproducing correct nucleotide base pairing when used in plain MD simulations (*Galindo-Murillo et al., 2016*). We hence conclude that the 'soft' (global) fitting potential (*Figure 1*) with a well-adjusted weight is a substantial factor in maintaining the modelled structure's accuracy, even in cases where neither the resolution nor the structural knowledge (e.g. a force field) are satisfactory.

Finally, the convergence of our and the alternative methods was assessed using the same distant starting structure (*Figure 8a*). Stereochemical quality statistics for the generated models are summarized in *Table 5*. Among all the methods tested, only CDMD was able to achieve a model accuracy

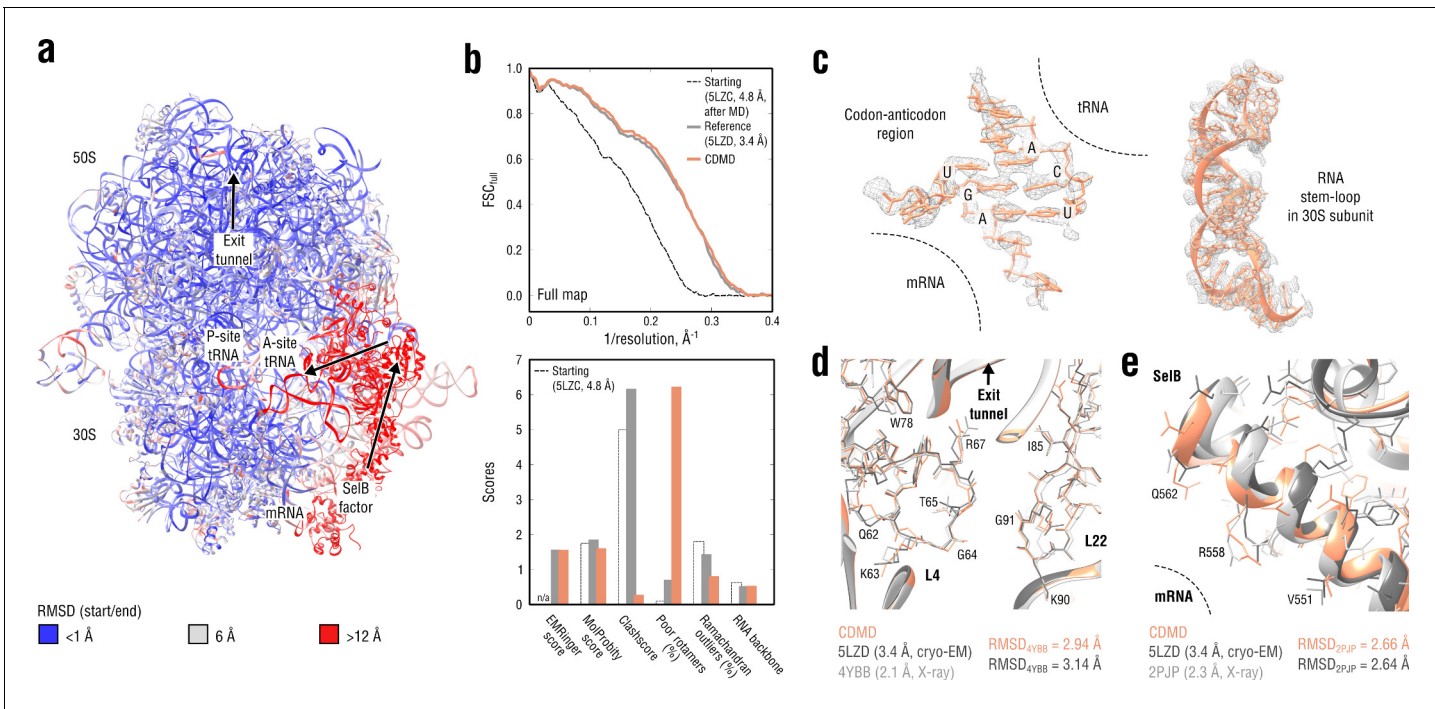

**Figure 9.** Refinement of a ribosome complex in the CR state into a 3.4 Å map of the GA state. (**a**) RMSD (RNA and protein backbone) between the starting (CR) and the final (GA) model showing the extent of rearrangements during the refinement. (**b**) Reciprocal-space agreement of the starting (black dashed), the reference (gray) and our model (orange) with the full map (top) and stereochemical quality for the three models assessed by EMRinger and MolProbity (bottom). (**c**) Representative regions showing local agreement between our model and the map (the codon-anticodon region and a RNA stem-loop in the 30S subunit). (**d, e**) Representative regions in the ribosomal exit tunnel (constriction site formed by L4 and L22 protein chains is shown) and in the SelB-mRNA contact interface, both demonstrating the closeness of our model to higher resolution control structures in terms of RMSD. Some protein residues are explicitly labeled.

DOI: https://doi.org/10.7554/eLife.43542.019

The following video and figure supplement are available for figure 9:

**Figure supplement 1.** Extension of *Figure 9* showing the comparison of the radii of convergence across different refinement methods for the ribosome system and using the same distant starting structure.

DOI: https://doi.org/10.7554/eLife.43542.020

**Figure 9—video 1.** Refinement trajectory for the 70S ribosome.

DOI: https://doi.org/10.7554/eLife.43542.021

and map-model agreement similar to the deposited reference. Although the histone part was refined similarly well by all of the methods, the DNA part was, if anything, only poorly refined (Refmac) and in most cases disintegrated (Phenix and Rosetta). The poor DNA quality of the Rosetta model is likely due to the internal Rosetta energy function that was originally designed for protein modeling (*Alford et al., 2017*), and a generalization for RNA/DNA is currently being developed. The fact that the Phenix refinement, in which DNA restraints were constantly switched on, was unable to converge to the reference structure might indicate that the starting structure is out of this method's radius of convergence. For Refmac, the optimal geometry weights necessary to obtain good agreement for the DNA and the histone part differed by 2–3 orders of magnitude such that a refinement of the entire nucleosome structure in a single run was impossible without severe geometry violations.

## Ribosome: refinement of a large protein-RNA complex

As a second example of a mixed nucleotide-protein system, we refined a large bacterial 70S ribosome/SelB complex in the codon reading (CR state; PDB ID: 5LZC) into the recently published 3.4 Å map (EMD ID: 4124, PDB ID: 5LZD; *Fischer et al., 2016*) of the GTPase-activated state (GA). Similar to the nucleosome test case, the chosen ribosome system is a nucleic-acid-protein complex (here, RNA) which undergoes non-trivial rearrangements in switching from CR to GA to form the codon-anticodon base pairs (local deviations reaching ~19 Å in the A-site tRNA and SelB elongation factor; *Figure 9a* and *Figure 9—video 1*). Besides these challenges, the refinement of such a large complex is a multiscale problem where local errors introduced by local adjustments accumulate and may propagate far beyond the refined location due to the large system size, if not resolved globally. With currently available refinement tools, several algorithms need to be applied iteratively to obtain a good quality model (*Fischer et al., 2016*). The ribosome refinement case, therefore, served to test if the method is able to handle local and global structural changes simultaneously while gradually improving the map-model agreement. We validated our results using the high-resolution X-ray structure of an *Escherichia coli* ribosome (PDB ID: 4YBB at 2.1 Å; *Noeske et al., 2015*) and the high-resolution X-ray structure of a SelB mRNA-binding domain (PDB ID: 2PJP at 2.3 Å; *Soler et al., 2007*).

*Figure 9b* compares our model with the reference structure in terms of map-model agreement and geometry. *Figure 9c* additionally shows local agreement of our model with the full map. Both models agreed well with the full map and with each other (RMSD $\approx$ 1.3 Å for all non-hydrogen atoms). The geometry scores for both models were largely consistent, except for the better Ramachandran statistics and clashscore in our model *vs.* the lower number of rotamer outliers in the deposited reference. Although this refinement case was not as challenging in terms of radius of convergence as the above examples (RMSD$_{ref}$ $\approx$ 2.5 Å for the starting structure after equilibration),

**Table 6.** Refinement statistics for the ribosome system.

|  | CDMD | Phenix | Rosetta | Refmac |
|---|---|---|---|---|
| FSCavg (full map) | 0.766 | 0.807 | 0.549 | 0.724 |
| EMRinger | 1.56 | 1.87 | 0.24 | 0.50 |
| Bond lengths (Å) | 0.017 | 0.016 | 0.062 | 0.010 |
| Bond angles (°) | 2.01 | 1.41 | 6.88 | 1.83 |
| MolProbity | 1.61 | 1.42 | 3.17 | 2.69 |
| All-atom clashscore | 0.28 | 7.65 | 152.9 | 9.34 |
| Ramachandran statistics: |  |  |  |  |
| Favored (%) | 94.09 | 99.25 | 90.84 | 92.06 |
| Allowed (%) | 5.10 | 0.75 | 5.78 | 6.69 |
| Outliers (%) | 0.81 | 0.0 | 3.38 | 1.25 |
| Poor rotamers (%) | 6.23 | 1.01 | 0.33 | 8.69 |
| CaBLAM flagged (%) | 14.0 | 19.0 | 23.7 | 16.8 |
| RNA backbone | 0.54 | 0.41 | 0.381 | 0.38 |

DOI: https://doi.org/10.7554/eLife.43542.034

CDMD was able to produce a good quality model despite the system size and the straightforwardness of the CDMD refinement, compared to the multi-method approach used in the original study (*Fischer et al., 2016*). We additionally verified our model's quality by comparing it and the reference with the higher resolution control structures (*Figure 9d,e*). As the control *Escherichia coli* ribosome structure (4YBB) was in a different translation state, we extracted only the 23S rRNA together with proteins L4 and L22 of the large 50S subunit (also shown in part in *Figure 9d*). From the control structure of the SelB mRNA-binding domain (2PJP) only the protein part was extracted (*Figure 9e*). Both reference and CDMD model were in good agreement with the 23S control structure showing RMSD values of ~3 Å, which is small given the size of the compared parts. The agreement was worse for the much smaller SelB part, where both models deviated from the control by ~2.7 Å in terms of RMSD. This can be explained by the lower local resolution of the cryo-EM density in this region. None of the compared models showed significantly smaller or larger $RMSD_{ref}$ values from the controls, suggesting that both models are equally good approximations of the 'true structure'.

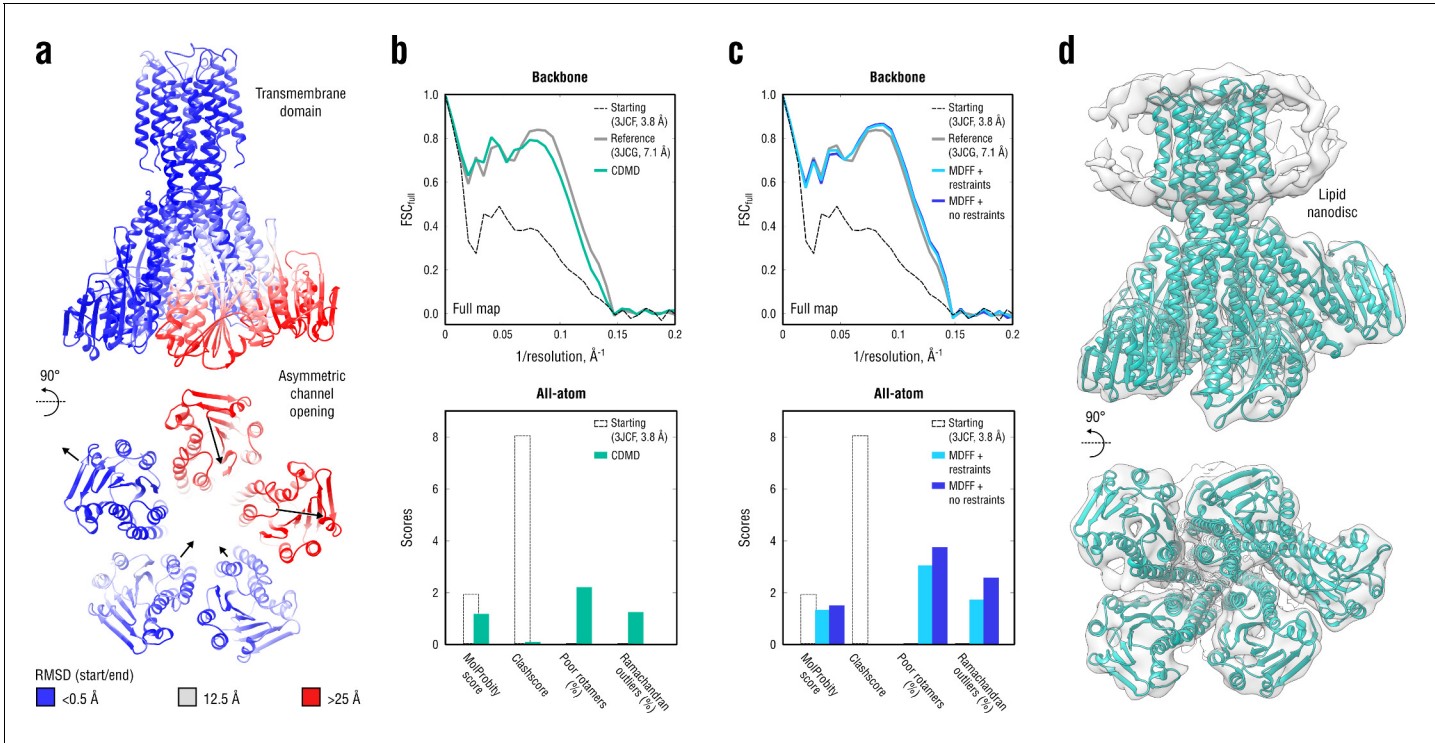

**Figure 10.** Refinement of a CorA magnesium transporter in the symmetric closed state into a low-resolution 7.1 Å map of the asymmetric open state. (a) RMSD ($C_\alpha$ atoms) between the starting (closed) and the reference (open) model showing the extent of rearrangements in the cytosolic part of the channel during refinement. (b) Reciprocal-space agreement of the starting (black dashed), the reference (gray) and our model refined with $k = 1.0 \times 10^5$ kJ mol$^{-1}$ (sea green) with the full map (top) and stereochemical quality for the three models assessed by EMRinger and MolProbity (bottom). The FSC curves were calculated using the backbone atoms only, whereas the full-atom models were used for geometry analysis. No EMRinger scores were calculated. (c) Same as in (b) but for the two MDFF models refined with (blue) and without (dark blue) secondary structure, chirality and *cis* peptide bond restraints. (d) Overlay of our full-atom model (see green) with full map.
DOI: https://doi.org/10.7554/eLife.43542.022

The following figure supplements are available for figure 10:

**Figure supplement 1.** Extension of *Figure 10* showing map-model agreement *vs.* rotamer or Ramachandran outliers for all CDMD and MDFF refinements.
DOI: https://doi.org/10.7554/eLife.43542.023

**Figure supplement 2.** Extension of *Figure 10* showing the structure of the gating pore.
DOI: https://doi.org/10.7554/eLife.43542.024

**Figure supplement 3.** Extension of *Figure 10* showing how the pore radius changes along the nonlinear gating pathway showin in *Figure 10—figure supplement 2* (bottom).
DOI: https://doi.org/10.7554/eLife.43542.025

To assess the radius of convergence of CDMD and the other alternative methods, we used the same starting structure as in *Figure 9a* (see *Figure 9—figure supplement 1*). Refinement statistics for the generated models are summarized in *Table 6*. Similarly to the NSF case, none of the tested alternative methods was able to achieve a map-model agreement and accuracy consistent with the reference or CDMD model. While Phenix performed relatively well in the ribosomal core domains, yielding a structure with good geometry, its convergence was poor for solvent-accessible parts and for the SelB factor (the region with the largest $RMSD_{ref}$ in *Figure 9a*). As a result, the Phenix model had a larger RMSD from the reference than the starting structure itself. Refinement with Rosetta yielded a rather poor model due to its energy function's unsuitability for DNA/RNA modeling. Also in this case the final $RMSD_{ref}$ value was larger than the starting one. Refmac was the only alternative method tested that refined the starting structure toward the reference, resulting in a final $RMSD_{ref}$ being smaller than the starting one. However, as in the Phenix refinement, it was unable to refine regions of the starting model that deviated strongly from the reference. Overall, these comparative modeling results confirm the necessity for a multi-method method approach when refining large systems such as the ribosome with current non-MD methods (*Fischer et al., 2016*). On the other hand, they also suggest that CDMD may drastically reduce the effort in such a refinement.

## CorA magnesium transporter: low-resolution map fitting

Having shown that CDMD provides plausible model geometries consistent with the map even when the cryo-EM density contains only partial structural information, we proceeded to a more challenging case where only low-resolution information was available. To this end, we refined the medium-resolution structure (PDB ID: 3JCF; *Matthies et al., 2016*) of a magnesium transporter CorA in the closed, symmetric $Mg^{2+}$-bound state into a 7.1 Å subclass map (EMD ID: 6552; PDB ID: 3JCG, backbone model only) of one of the $Mg^{2+}$-free, asymmetric open states. Unlike in the above cases, the starting structure was equilibrated in explicit solvent but not further subjected to long-term MD at $T = 300K$ as the RMSD from the target state already reached ~25 Å for some parts of the channel (*Figure 10a*). A series of full map refinements was performed using a constant $\sigma$ value of 0.6 nm and a set of low force constants (0.5, 1.0, 1.5 and 2.0 $\times 10^5$ kJ mol$^{-1}$), which was consistent with the low map resolution. We found the values between 1.0 and 2.0 $\times 10^5$ kJ mol$^{-1}$ to be optimal for the refinement because they enforced sufficient map-model agreement while preserving secondary structure and maintaining reasonable model geometry. Increasing the force constant beyond 2.0 $\times 10^5$ kJ mol$^{-1}$ did not result in better map-model agreement, and for much higher values (comparable with those in, *e.g.*, *Figure 3*; data not shown), the model quality dropped significantly, suggesting overfitting of the map. We additionally compared our results with CorA models produced by MDFF, a well-established tool originally designed for exactly such refinement cases (see Materials and methods for the MDFF protocol).

**Table 7.** Refinement statistics for the CorA system.

|  | CDMD | MDFF (restraints) | MDFF (no restraints) |
| --- | --- | --- | --- |
| FSCavg (full map) | 0.673 | 0.711 | 0.713 |
| Bond lengths (Å) | 0.017 | 0.021 | 0.020 |
| Bond angles (°) | 2.11 | 2.34 | 2.30 |
| MolProbity | 1.15 | 1.31 | 1.48 |
| All-atom clashscore | 0.07 | 0.0 | 0.0 |
| Ramachandran statistics: |  |  |  |
| Favored (%) | 94.77 | 93.01 | 90.09 |
| Allowed (%) | 4.01 | 5.29 | 7.36 |
| Outliers (%) | 1.22 | 1.70 | 2.55 |
| Poor rotamers (%) | 2.18 | 3.02 | 3.72 |
| CaBLAM flagged (%) | 11.1 | 13.3 | 18.1 |

DOI: https://doi.org/10.7554/eLife.43542.035

A direct comparison to the deposited reference (3JCG) was complicated by the fact that all side chains were removed in the original study due to the lack of density to assign their conformations (*Matthies et al., 2016*). Therefore, all side chains in our and the MDFF models were exempted from the FSC curve calculations (*Figure 10b,c*, top), but we used the all-atom models to assess the stereochemistry (*Figure 10b,c*, bottom). *Figure 10d* additionally shows an overlay of our model with the map. Both our model and the one refined with MDFF correlated similarly well with the low-resolution map, with the latter correlating better at spatial frequencies above 0.1 Å$^{-1}$ (irrespective of whether or not secondary structure restraints were enabled). However, the quality of the MDFF models was systematically lower than that of our models (*Figure 10b,c*, bottom and *Figure 10—figure supplement 1*), although each model had almost zero steric clashes (due to the use of an atomistic force field). The quality of the MDFF model decreased as the restraints were released (*Figure 10c* and *Figure 10—figure supplement 1*, blue *vs.* dark blue). Stereochemical quality statistics for all CDMD and MDFF models shown in *Figure 10* are summarized in *Table 7*.

Does the CorA model refined with CDMD represent the asymmetric open conformation more accurately? To answer this question, a comparison with high-resolution structures of the asymmetric open state is required which were not available for the CorA system. Nevertheless, there is experimental and simulation evidence that removal of the regulatory Mg$^{2+}$ ions from the symmetric CorA conformation leads to large-scale rearrangements of the intracellular stalk helices that increase the diameter of the gating pore, hence triggering ion conduction (*Chakrabarti et al., 2010*; *Dalmas et al., 2014*). Although pore opening is hard to infer by inspecting the cryo-EM map due to its low resolution, it is conceivable that an accurate refinement of the symmetric CorA structure with well-balanced contributions from both cryo-EM map and molecular force field would induce such structural changes. We therefore compared the structure of the gating pathway in all of the models analyzed in *Figure 10b,c*. Specifically, we analyzed the structure and biophysical properties of the CorA gating pathway using the CHAP program (*Rao et al., 2017*; *Klesse, 2019*). This analysis yielded a surface representation of the gating pathway as well as its radius at every position along the pathway (*Figure 10—figure supplement 2* and *Figure 10—figure supplement 3*). The fact that we used explicit solvent in all of the simulations also allowed us to estimate the solvent density in the pore. The gating pore of our CorA model was, on average, wider along the pathway than that of the closed symmetric structure. The average solvent density in the gating pore of our CorA model was lower, consistent with the increased pore volume. Surprisingly, the gating pore in the MDFF model was rather curled and significantly smaller in volume than that of the closed symmetric structure, despite the intracellular stalk helices adopting an asymmetric conformation very similar to our model. It was mainly due to the gate closure in the central part of the transmembrane domain (pore radii reaching 0.5 Å, see *Figure 10—figure supplement 3*) that led to a complete permeation block as indicated by zero number density in this region (no water molecules were present). We speculate that this conformation of the gating pore in the MDFF model would lead to even weaker conductance than in the case of the closed symmetric channel in free MD simulations. Although a higher-resolution control is still required to validate the open structure produced by CDMD, the fact that the gating pore opens and is on average more solvated in the CDMD fitting simulations is in agreement with previous biochemical and simulation data (*Chakrabarti et al., 2010*; *Dalmas et al., 2014*).

Remarkably, MDFF was unable to induce gate opening. To test whether it is the secondary structure restraints (an option strongly recommended for low-resolution refinement with MDFF) that prevented the torque generated by rearranging the intracellular domains from propagating into the gate region, we switched off all the restraints and re-refined the starting structure. This restraint-free MDFF setup did not lead to gate opening either (*Figure 10—figure supplement 2*, rightmost structure; *Figure 10—figure supplement 3*, bottom right plot). We note that the restraint-free MDFF setup used here was identical to our refinement setup, except for the biasing potential form. We therefore speculate that it is the 'hard' MDFF biasing potential (each density grid point is an attractor) that makes the refined model less compliant and restricted only to the highest-density regions. On the contrary, the 'soft' correlation-based potential (global map-model overlap is maximized) allows more internal flexibility while maintaining a comparable degree of map-model agreement (*Figure 10b,c*, top, and *Figure 10—figure supplement 1*). When coupled to a stereochemically accurate treatment (atomistic force field), our refinement protocol might thus be useful in modeling biologically relevant transitions.

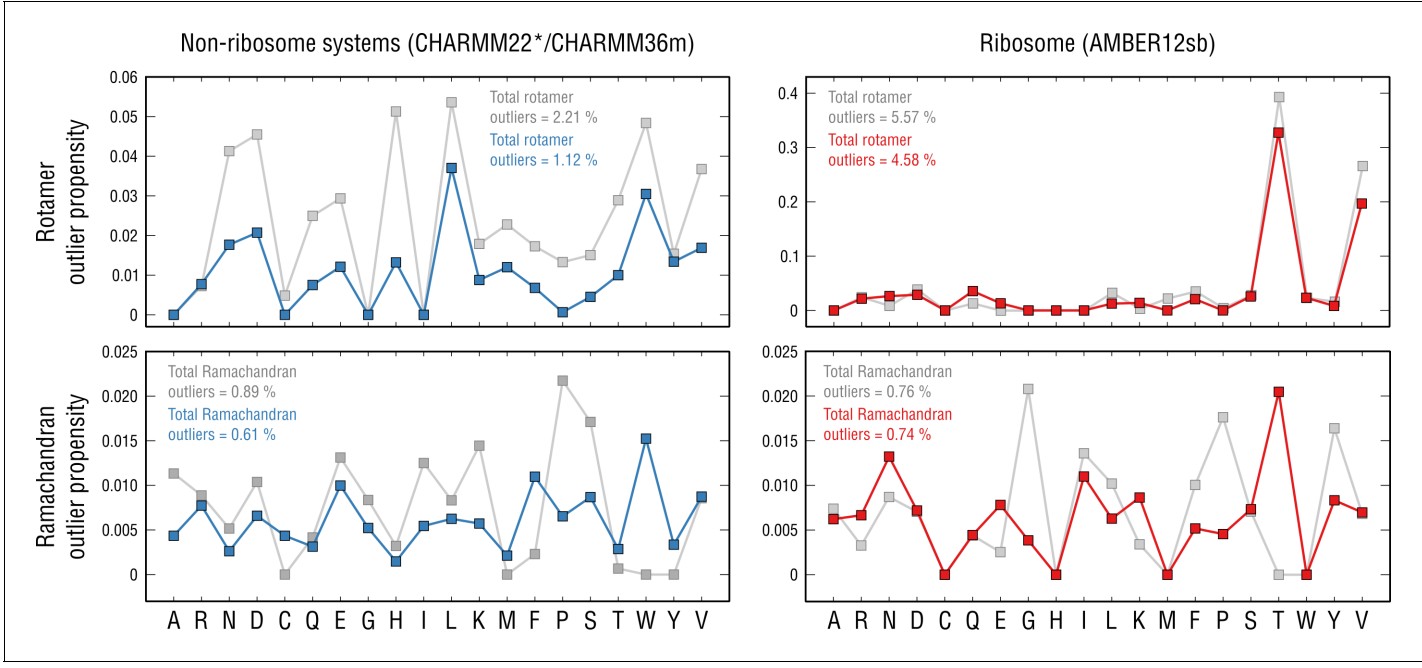

**Figure 11.** Per-residue outlier analysis. Per-residue rotamer and Ramachandran outlier propensities calculated from the non-ribosome structures (left panels; CHARMM22*/CHARMM36m force fields were used (*Piana et al., 2011*; *Huang et al., 2017*) and the ribosome structure (right panels; AMBER12sb was used; *Lindorff-Larsen et al., 2010*) after pre-equilibration with the force field (gray dots) and after the actual refinement with our approach (blue and red dots, respectively).

DOI: https://doi.org/10.7554/eLife.43542.026

## Force field influence and model accuracy

We now return to the question of why in many of the above test cases the remarkable reduction of steric clashes by CDMD often comes at the cost of increased rotamer and/or Ramachandran outliers, as compared to the deposited reference models (*Figure 3*, *Figure 4*, *Figure 6a,b*, *Figure 7*, *Figure 9*). It should be first noted that, by construction, the refinement methods used to produce the deposited models (COOT, Phenix, Rosetta, Refmac) often enforce accurate rotamer and Ramachandran distributions, whereas, also by construction, force-field-based refinement guarantees almost zero steric clashes. The question therefore narrows down to whether the structural errors observed in our models (a) arise due to imperfections of the force fields in use (*e.g.*, inaccurate backbone dihedral, rotamer torsion angles or atomic partial charges), (b) are enforced by the cryo-EM data (real or noise), or (c) are a synergetic combination of these effects. It is conceivable that in scenario (a), outliers of certain amino acid types would be systematically and significantly overrepresented in the total outlier pool, both before and after structure refinement. In contrast, a low to moderate number of outliers being rather homogeneously distributed across the amino acid alphabet would point to scenario (b). These unspecific outliers might, in turn, be contributed by poorly determined map regions or be naturally occurring outliers in densely packed protein interiors.

To test this idea, we performed a per-residue analysis of the rotamer and Ramachandran outliers detected in our models (*Figure 11*, left; see Materials and methods). For each of the above non-ribosome test cases, the final structures and the ones pre-equilibrated in explicit solvent prior to refinement were analyzed, and the resulting statistics were merged. The ribosome case was analyzed separately because a different force field family was used (*Figure 11*, right). We first asked if our refinement procedure (*Figure 1* and *Figure 2*) per se improves the rotamer and Ramachandran distributions. Indeed, there was an overall decrease both in the average per-residue outlier propensity and in the total outlier occurrence (*Figure 11*, gray *vs.* blue or red dots), irrespective of the force field. These data indicate that a considerable fraction of structural conflicts is resolved by the refinement and support scenario (b) above: that the cryo-EM density improves the geometry of initially poorer models leaving a certain number of residue-unspecific outliers.

However, mainly for the rotamer distributions, we observed that those peaked at certain amino acid types more frequently, both before and after structure refinement. In particular, for the non-ribosome systems refined using CHARMM22*/CHARMM36m (*Piana et al., 2011*; *Huang et al., 2017*), aspartates, leucines, tryptophans and valines contributed most to the overall outlier occurrences, which, however, remained rather low (outlier propensities of ~0.04 or lower). AMBER12sb (*Lindorff-Larsen et al., 2010*) force field used for the ribosome refinement performed similarly to CHARMM22*/CHARMM36m for all rotamer types except for threonines and valines which systematically disagreed with their knowledge-based conformations (an order of magnitude higher outlier propensities, ~0.33 and ~0.2, respectively). The presence of systematic structure errors, therefore, supports scenario (a) as these errors are residue- and force-field-specific and not corrected by the refinement.

Thus, there are several lines of evidence that the structural errors detected in our models are likely to be a combination of cryo-EM data-specific outliers and force field artifacts (scenario (c) above). It should be noted that the quality of a refined structure is only as good as the experimentally determined map. Low determinacy of 3D classification and poor alignment of cryo-EM particles may be related to the non-ideal nature of structure refinement. Furthermore, conformational heterogeneity and ensemble averaging misinterpreted as a single structure can be another source of structural errors in which case an ensemble refinement may be required (*Rice et al., 1998*; *Burnley et al., 2012*; *Herzik et al., 2019*; *Bonomi et al., 2018*). In our present study, we have concentrated on single-structure refinement and have implicitly assumed that each map stems from a unique conformational state of the biological complex in question. This is a good approximation for high-resolution maps (e.g. *Figure 3*). For medium- and low-resolution maps, the best representation might not be a single structure but rather a weighted set of structures. The TRPV1 system (*Figure 5*) is one example of such a case in our study. Here, three independent refinements produced three structures with very similar conformations in the high-resolution TM region and with much more variability in the low-resolution ARD domains (*Figure 5—figure supplement 2*). The subject of ensemble refinement, however, lies beyond the scope of this study.

Correction of the force-field outliers, in turn, is tightly linked to the general force field development. Still undefined, apparently, is the way a reduction of steric clashes should be reconciled with an improvement of other stereochemical measures (e.g. rotamer and Ramachandran statistics, $C_\beta$ deviations, chirality errors, etc.), as these might currently represent conflicting optimization goals. On the one hand, it is well known what the expectations are for such errors (*Bower et al., 1997*; *Shapovalov and Dunbrack, 2011*; *Renfrew et al., 2008*; *Towse et al., 2016*). On the other hand, incorporation of this knowledge into the force field development is challenging as it fundamentally differs from the way a rotameric library is constructed (*Towse et al., 2016*). Even without solutions

**Table 8.** Data sets used in this study.

| System | Resolution, Å | EMDB | Deposited PDB | Method | Citation |
|---|---|---|---|---|---|
| Aldolase | 2.6 | 8743 | 5VY3 | Rosetta/Phenix | *Herzik et al. (2017)* |
| Tubulin | 4.1 | n/a* | 3JAS, 6DPV | COOT/Refmac | *Zhang and Nogales (2015)* |
| TRPV1 | 3.3 (2.5–7) | 5778 | 3J5P | COOT | *Liao et al. (2013)* |
| TRPV1 (TM) | <3.3† | 5778 | 3J9J | Rosetta | *Barad et al. (2015)* |
| TRPV1 (mono) | 3.3 (2.5–7) | 5778 | n/a‡ | ReMDFF | *Wang et al. (2018)* |
| NSF | 3.9 | 9102 | 6MDO | COOT/Phenix§ | *White et al. (2018)* |
| Nucleosome | 3.7 | 3947 | 6ESF | COOT/Phenix | *Bilokapic et al. (2018)* |
| CorA | 7.1 | 6552 | 3JCG | COOT/Phenix | *Matthies et al. (2016)* |
| 70S Ribosome | 3.4 | 4124 | 5LZD | COOT/Rosetta/Phenix | *Fischer et al. (2016)* |

*Map segment derived from an asymmetric, 14-protofilament, kinesin-decorated microtubule reconstruction (provided by courtesy of R. Zhang).

†TM region of TRPV1 has a much higher resolution than ARDs.

‡Provided by courtesy of S. Wang.

§Optimized Phenix protocol was used (*Afonine et al., 2018*; *White et al., 2018*) that differed from that used for the other systems (*Adams et al., 2010*).

DOI: https://doi.org/10.7554/eLife.43542.027

to the above problems, our results clearly imply that the correlation-driven MD refinement is able to generate models with a quality consistent with well-established methods given the map quality.

## Conclusions

We have established and validated the accuracy and convergence of CDMD by applying it to a diverse set of test cases (*Figure 3*-*Figure 10*) covering the most typical refinement challenges posed by modern cryo-EM: starting models in conformational states distant from the target map, poor starting model geometry, highly heterogeneous local map resolution, multiscale fitting where local adjustments have to be balanced by global optimization, low- or medium-resolution maps where high-resolution features have to be drawn elsewhere, or combinations thereof. Our results demonstrate that all these challenges can be handled by CDMD without additional backbone- and rotamer-specific restraints or manual intervention. Yet, our method has a much higher radius of convergence than current non-MD methods, which is largely independent of the map resolution and which is comparable to that of the most recent MDFF versions (*Singharoy et al., 2016*; *Wang et al., 2018*), namely, at least 6–7 Å in terms of the overall RMSD from the target state and at least 25 Å in terms of local deviations. We, once again, emphasize that some of the tested alternative approaches are not entirely optimal for the presented refinement cases, mainly due to the starting structures we used being too distant and/or containing RNA/DNA. The comparisons with the alternative methods, however, clearly illustrate the challenge of reconciling local refinement accuracy and large convergence radius, in which a proper treatment of concerted molecular motions plays a major role. Finally, the CorA case demonstrates that CDMD is capable of capturing biologically relevant conformational changes, thus underlining the advantage of using the correlation-based biasing potential for low-resolution refinements.

The dramatic recent advances in image collection and processing have resulted in cryo-EM densities being generated at an unprecedented pace, with the above challenges tightly interlaced. CDMD largely overcomes the shortcomings of the refinement methods available so far, namely, the poor convergence and accuracy when applied to maps with medium/low resolution (non-MD methods) or high resolution (MD-based flexible methods). Integrating well-established techniques, for example, the correlation biasing potential (*Orzechowski and Tama, 2008*) or simulated annealing (*Brünger et al., 1987*; *Brunger and Adams, 2002*), into an optimized protocol involving a chemically accurate force field, adaptive resolution and the half-map-based validation scheme has finally allowed to overcome these shortcomings in an automated fashion. We note that our method can still be used for quick (re-)refinement of close starting models against high-resolution maps (for example, *Figure 6a,b*) or as a tool for flexible fitting of distant atomic models into low-resolution maps (*Figure 10*), which underlines its flexibility and universality. Also, our implementation based on the highly scalable, GPU-accelerated GROMACS engine provides a good trade-off between refinement time and hardware cost (see the benchmark section in Materials and methods). In combination with modern de novo chain building tools, it provides a fully automated and human-bias-free framework for quantitative interpretation of modern cryo-EM data.

## Materials and methods

### Preparing maps and starting structures

Unless differently specified, VMD (*Humphrey et al., 1996*) and UCSF Chimera (*Pettersen et al., 2004*) were used to perform all map and structure manipulations. We used the previously published data sets specified in *Table 8*. All maps used in this study and the corresponding deposited reference models were publicly available for download from the EMDB or kindly provided by the corresponding authors.

The maps were trimmed using an orthorhombic box to reduce the system size for further simulation. The trimmed maps were then converted into the CCP4 format for GROMACS compatibility using the *em2em* software (http://www.imagescience.de/em2em.html). X-ray or cryo-EM structures of an aldolase (PDB ID: 6ALD; *Choi et al., 1999*), a GDP-bound tubulin after ~3 µs of MD simulation (PDB ID: 4ZOL; *Wang et al., 2016b*; *Igaev and Grubmüller, 2018*), a TRPV1 channel (PDB ID: 3J5P; *Liao et al., 2013*), a NSF complex (PDB ID: 3J94; *Zhao et al., 2015*), a nucleosome (PDB ID: 6ESI; *Bilokapic et al., 2018*), a CorA channel in the symmetric state (PDB ID: 3JCF; *Matthies et al.,*

*2016*), and a 70S ribosome in the CR state (PDB ID: 5LZC; *Fischer et al., 2016*) were used as starting models.

The aldolase and tubulin starting structures were prepared as described in *Herzik et al. (2017)* and *Igaev and Grubmüller (2018)*, respectively. For the TRPV1 starting structure, the missing non-terminal loops (residues 503–507) were modeled in using MODELLER version 9.17 (*Fiser et al., 2000*). The regions with unassigned side chains (UNK residues 720–763) were substituted by those from PDB ID: 5IRZ (*Gao et al., 2016*). All missing non-terminal residues in the starting NSF structure were modelled in using MODELLER. Two of four ATP molecules (chains A and E) were turned into ADP, consistent with the reference (PDB ID: 6MDO). The SNAP-25A 17 N-terminal residues were copied from PDB ID: 6MDP (*White et al., 2018*) and rigid-body fitted into the density. In the nucleosome starting model (PDB ID: 6ESI), a part of DNA was missing due to unwinding. This part, as well as short terminal tails of the histone complex, were copied from PDB ID: 3LZ1 (*Vasudevan et al., 2010*) to match the chain sequences in the deposited reference model (PDB ID: 6ESF). For the CorA starting structure, the $Mg^{2+}$ ions and the N-terminal tails (residues 1–18 are not reflected in the density) were removed. The preparation of the 70S ribosome starting model is described in a separate section below.

We emphasize that having a complete starting model is not a requirement for our method, and that models with missing regions can still be refined. The starting model completion as described above was done to match the structure of the deposited models and to facilitate a fair comparison. Parts of the models for which there was no clear experimental density (e.g. residues 38–46 in tubulin's $\alpha$-subunit or residues 111–198 of TRPV1) were not subjected to the density fitting potential and removed later on during structure analysis. It is however preferable to have complete models if plain MD simulations are to be run with the refined structures.

## Simulation setup and refinement of non-ribosomal systems

The non-ribosomal simulation systems consisted of the atomic model pre-aligned with the full map in a triclinic box of TIP3P water molecules (CHARMM-modified) with 1.5–2 nm padding and 0.15 M KCl. All bond lengths were constrained with the LINCS algorithm and virtual sites were used for hydrogens, allowing a 4-fs integration time step. CHARMM36m force field (*Huang et al., 2017*) was used in the CorA refinement for consistency with MDFF, while CHARMM22* force field (*Piana et al., 2011*) was used in all other cases. The simulation parameters are described elsewhere (*Igaev and Grubmüller, 2018*). All files as well as a step-by-step guide necessary for setting up the aldolase refinement (*Figure 3*) are available at https://www.mpibpc.mpg.de/grubmueller/densityfitting.

The starting models were subjected to the above MD setup followed by steepest-descent energy minimization, a short equilibration in NVT ensemble at $T$ = 100 K with position restraints on the solutes' heavy atoms (spring constant $k_{posres}$ = 1000 kJ mol$^{-1}$ nm$^{-1}$, simulation time 1 ns), and a 50-ns equilibration in NPT ensemble at $T$ = 300 K to relax the systems and to increase the deviation from the target state before refinement against one of the half-maps (training map) as sketched in *Figure 2—figure supplement 1*. Structures with the largest deviation from the deposited reference models were selected, and continuous refinement against the training map was applied in three different incarnations: (i) $T$ = 100 K for 50 ns, (ii) $T$ = 150 K for 30 ns, and (iii) $T$ = 200 K for 20 ns. Here, higher temperature protocols facilitated thermodynamic sampling, allowing the system to reach good map-model agreement within shorter simulations times. However, we observed that lower temperature protocols were less prone to overfitting when applied to very noisy half-maps or in combination with unusually high force constants. If neither of these conditions was true, all protocols performed similarly well.

The starting values for $\sigma$ and $k$ were set to 0.6 nm and $0.5 \times 10^5$ kJ mol$^{-1}$. After 3 ns of refinement with $\sigma_{start}$ and $k_{start}$, both parameters were linearly ramped to their target values while keeping the pressure at 1 atm. The target value $\sigma_{stop}$ was determined by correlating the deposited reference model with the respective half- and full map at $\sigma$ values ranging from 0.6 to 0.1 nm with a 0.01 nm increment, and the value (usually 0.15–0.25 nm) yielding the highest correlation was taken as $\sigma_{stop}$. Except in the cases of CorA and NSF, target values of $k_{stop}$ between 3 and $5 \times 10^5$ kJ mol$^{-1}$ were optimal, as suggested by the half-map cross-validation. These values were found to not cause overfitting for the given set of test systems and yet to be sufficiently strong to guarantee good map-model agreement. We do not recommend using higher $k_{stop}$ values for systems of this size and

resolution ($10–40 \times 10^3$ atoms, high to medium resolution) as those could cause frequent numerical instabilities due to the density-based forces overriding those from the atomistic force field. For the CorA channel, we used $\sigma = 0.6$ nm and $k = 1 \times 10^5$ kJ mol$^{-1}$ throughout the refinement simulation against the full map because no high-resolution features were available and only large-scale concerted motions had to be accomplished. For the NSF system, a wide range of target force constants ($2–8 \times 10^5$ kJ mol$^{-1}$) was tested to identify the optimal density bias (described later in Materials and methods).

After cross-validation with the second half-map (discussed later in Materials and methods), a short 20 ns refinement run was performed with the full reconstruction in order to account for high-resolution features not present in both half-maps (*Figure 2—figure supplement 1*). It consisted of a 15 ns simulated annealing step and additional 5 ns to obtain the final refined set of structures, while $\sigma_{\text{stop}}$ and $k_{\text{stop}}$ were kept constant. Two different annealing schemes were tested: rapid heating of the system to 300 K and 500 K within 1 ns and 3 ns, respectively, followed by a slow cooling down phase. No significant advantage of one scheme over the other was observed. Extending the cooling phase to 20 ns did not have any further improving effect either. Hence, we used the 300 K annealing scheme for all systems.

## Simulation setup and refinement of 70S ribosome

The structure of the 70S ribosome in a distant conformational state (codon reading, PDB ID: 5LZC) was used as a starting model. Additional Mg$^{2+}$ ions were placed according to their positions in the structure of the GTPase-activated state (*Fischer et al., 2016*). The structure was protonated, solvated, and ions were added as described previously (*Arenz et al., 2016*). The system, was then pre-equilibrated in three steps: first, steepest-descent energy minimization; second, equilibration of the solvent with position restraints on non-solvent heavy atoms (spring constant $k_{\text{posres}} = 1000$ kJ mol$^{-1}$ nm$^{-1}$, simulation time 50 ns); and finally, linear decrease of the position restraints to zero during 20 ns. For the refinement, the waters and ions within a distance of 25 Å from any solute atom of the pre-equilibrated structure were kept, resulting in a water-ion shell around the ribosome. Additional solvent and ions were introduced to fill a triclinic box with ~1.2 nm padding around the solute. The AMBERFF12sb force field was used (*Lindorff-Larsen et al., 2010*); further simulation parameters can be found elsewhere (*Fischer et al., 2016*).

The prepared ribosome system was first run at $T = 300$ K and $p = 1$ atm for 5 ns with position restraints on the ribosome's non-hydrogen atoms to equilibrate the extra solvent added during the preparation phase. The restraints were then released, the temperature was set to 100 K, and a refinement run was performed for 50 ns using the training map resampled on a sparser grid (reduced to $247 \times 247 \times 247$ grid points) to speedup calculations for the large-scale transitions (e.g. tRNA and mRNA or ribosomal platform motion). The parameters were linearly changed from $\sigma_{start} = 0.6$ nm and $k_{start} = 4.5 \times 10^5$ kJ mol$^{-1}$ to $\sigma_{stop} = 0.2$ nm and $k_{stop} = 3.6 \times 10^6$ kJ mol$^{-1}$, while keeping the temperature and pressure at 100 K and 1 atm, respectively. Such high values of the force constant were chosen given the particularly large system size and the different force field used (AMBERFF12sb *vs.* CHARMM22*/CHARMM36m) which we found to decrease the overall system compliance. We then performed a short refinement run on the original training map ($400 \times 400 \times 400$ grid points) for 5 ns to account for high-resolution features not present in the reduced training map. After cross-validation with the second half-map, the 70S ribosome was additionally refined against the full reconstruction (involving simulated annealing) as described in the previous section.

## Cross-validation and stereochemical quality

We used a half-map-based validation approach similar to that proposed for refinement in REFMAC (*Amunts et al., 2014*; *Brown et al., 2015*) and the recent automated Rosetta protocol (*Wang et al., 2016a*). However, unlike in the Rosetta scheme, the cross-validation is used in our work to determine a range of $k$ values that the model can tolerate without being overfitted, but not to score the refined ensemble and preselect models for further refinement steps.

In practice, overfitting was assessed by monitoring training map-model and validation map-model FSC, FSC$_{\text{train}}$ and FSC$_{\text{val}}$, simultaneously every 5 ns of the half-map refinement. To this end, structure snapshots were extracted from the refinement trajectory and stripped of all waters, ions, hydrogens, and virtual sites. The periodic box was adapted to match that of the half-map, and the structures

were finally converted into simulated densities sampled on the same grid using a custom GROMACS tool gmx map and the respective $\sigma_{stop}$. If there was a strong divergence between $FSC_{train}$ and $FSC_{val}$ at the last stage of the half-map refinement – which we observed only for unusually high $k_{stop}$ values or very noisy half-maps – a structure at a lower force constant from the same trajectory that did not feature hallmarks of overfitting was used for the full map refinement (see *Figure 2—figure supplement 1*). The force constant was not further increased during the full map refinement as there was no longer a validation map to control overfitting.

The average structure from the last 5 ns of the full map refinement was used as the final model to calculate $FSC_{full}$ and for comparison with the deposited reference. All FSC calculations were done with the PDBe Fourier Shell Correlation Server (https://www.ebi.ac.uk/pdbe/emdb/validation/fsc/). The quality of the final average structure was assessed by EMRinger (*Barad et al., 2015*), MolProbity (*Chen et al., 2010*) and CaBLAM (*Richardson et al., 2018*) implemented in PHENIX version 1.14 (*Afonine et al., 2018*). X-ray scattering positions were consistently used to place hydrogen atoms with MolProbity (default option) for all models analyzed in this study. Using nuclear positions to place hydrogens led to a increase in the number of steric clashes by 15–20%. This number, however, never exceeded 0.4–0.5 for the models refined with our method or MDFF, as those had almost zero steric clashes by construction. Average FSC ($FSC_{avg}$) was calculated by integrating the FSC curves over the resolution shells starting from 1/100 Å$^{-1}$ to $1/r$ where $r$ is the full map resolution according to the 0.143 criterion (Equation 2 in *Brown et al., 2015*).

We note that the purpose of the half-map validation discribed above (see also *Figure 2—figure supplement 1*) is to find the optimal range of force constants that do not cause overfitting and yet guarantee good map-model agreement (e.g., in terms of FSC). For many cryo-EM reconstructions, half-maps are not always deposited, which, of course, does not preclude the use of our method. In such cases, it is possible to perform a series of independent refinements against the deposited full map using a range of target force constants and to select a final model(s) with the best map-model agreement and geometry (see the NSF refinement example). It is, however, strongly advised not to omit the half-map cross-validation because: (a) it is a computationally much less expensive strategy – one half-map refinement *vs.* 5–7 independent full map refinements per system, and (b) it gives more confidence in the final model's accuracy.

## Comparative refinement with alternative methods

Comparative refinement with Phenix, Rosetta, Refmac, and MDFF was performed using the most recent versions of the respective software packages. More specifically, we used the latest official Phenix release 1.14–3260 (*Afonine et al., 2018*) in combination with the recently proposed optimized refinement protocol (*White et al., 2018*). The Phenix refinement involved three steps with 15 cycles each: (a) rigid-body placement of individual chains, (b) refinement without secondary structure or Ramachandran restraints (global minimization, local grid search done in each cycle, simulated annealing done only once, and ADP refinement), and (c) refinement with secondary structure and Ramachandran restraints enabled. The *oldfield* target function was used, and different values of *plot_cutoff* (between 0.1 and 0.5) and *weight_scale* (between 0 and 10) were tested to obtain better structures in terms of $CC_{mask}$, Ramachandran outliers and CaBLAM scores. Non-crystallographic symmetry (NCS) restraints were enabled where possible.

To produce Rosetta models, we used the latest Rosetta build 2018.48.60516 in combination with the automated fitting protocol (*Wang et al., 2016a*) following the most recent density fitting tutorial (http://faculty.washington.edu/dimaio/files/rosetta_density_tutorial_aug18.pdf). For each case presented in the study, ~100 Rosetta models were independently refined against one of the half-maps and validated using the second half-map. The structure possessing the best scores (Rosetta, MolProbity, EMRinger and $|FSC_{train} - FSC_{val}|$) was further refined against the full map to produce the final Rosetta model. NCS restraints were enabled where possible. The density fitting weight was set to 35 except for the aldolase refinement, where it was set to 45 in accordance with the much higher map resolution.

Refmac5 from the latest version of the CCP4 suite (v. 7.0.067) optimized to work with cryo-EM densities (*Brown et al., 2015*; *Kovalevskiy et al., 2018*) was used to produce Refmac models. Jelly-body restraints on the current interatomic distances and local NCS restraints were enabled to improve the radius of convergence. To identify the optimal weight between geometry and experimental data, we performed multiple refinement runs with weights starting at 0.0001 and doubling at

every next run until the value of 0.1 was reached. The optimal weight was determined using the half-map validation (*Brown et al., 2015*) and by assessing the model geometry with MolProbity, which was typically on the order of 0.001–0.01 for protein systems.

NAMD 2.13 (*Phillips et al., 2005*) was compiled with Intel Compiler 18.0, Intel MPI 2018 and without CUDA support. The recent MDFF refinement protocol for low-resolution cryo-EM maps was adopted from *Gurel et al. (2017)*. Briefly, MDFF was performed in explicit solvent and in the presence of 150 mM KCl, using CHARMM36m parameters (*Huang et al., 2017*), and was run in three steps: 2000 steps of energy minimization with the fitting potential disabled, 10–50 ns of MD simulation at $T = 200$ K using a low force constant until the real-space correlation converged (several GSCALE values in the range between 0.05 and 0.5 were tested), followed by another round of energy minimization (2000 steps) with a much higher force constant (GSCALE = 10). Only backbone atoms were allowed to feel the fitting potential due to the lack of side chain density information, and all the other atoms were subjected only to molecular dynamics. We additionally tested two versions of the above protocol: with secondary structure, chirality and *cis* peptide bond restraints being enabled to minimize the risk of overfitting and without such restraints for consistency with our method. The optimal GSCALE value for the CorA system was ~0.1. Higher values led to strong structure distortions even in the presence of restraints. The value GSCALE = 0.05 was too low to induce the asymmetric channel opening.

## Per-residue outlier propensity

The outlier propensities shown in *Figure 11* were calculated using the following reasoning. We seek to calculate the probability that a particular residue is an outlier given its amino acid type, $P(out|aa)$. Invoking Bayes' theorem, we obtain:

$$P(out|aa) = \frac{P(aa|out)P(out)}{P(aa)}, \tag{1}$$

where $P(aa)$ reflects the amino acid occurrence probability, $P(out)$ represents the overall probability of outlier occurrence, and $P(aa|out)$ is the probability that an outlier has a particular amino acid type. Denoting the total number of amino acids and outliers as $N_{tot} = \sum N_{aa}$ and $n_{tot} = \sum n_{aa}$, respectively, where $N_{aa}$ and $n_{aa}$ are their per-residue components, we find that the sought per-residue propensity simply reflects the number of outliers for this particular residue type scaled according to its occurrence in the amino acid sequence, that is:

$$P(out|aa) = \frac{n_{aa}}{N_{aa}}. \tag{2}$$

**Table 9.** Performance of the refinement benchmarks for $\sigma = 0.2$ nm on two hardware configurations.

| System | D ($\times 10^6$) | N ($\times 10^6$) | W (ns/d) | S (ns/d) | $T_S$ (days, short) | $T_S$ (days, long) |
|---|---|---|---|---|---|---|
| Aldolase | 1.3 | 0.11 | 33.4 | 39.9 | 1.0 | 1.8 |
| Tubulin | 0.2 | 0.09 | 87.0 | 110.9 | 0.4 | 0.6 |
| TRPV1 | 1.9 | 0.37 | 16.7 | 21.5 | 2.0 | 3.3 |
| Nucleosome* | 0.6 | 0.17 | 20.7 | 29.9 | 1.3 | 2.3 |
| NSF | 1.3 | 0.36 | 18.5 | 24.9 | 1.6 | 2.8 |
| CorA | 1.2 | 0.31 | 25.4 | 29.7 | 1.3 | 2.4 |
| 70S Ribosome | 64.0 | 3.10 | 1.4 | 2.1 | 19.1 | n/a |

$D$ = number of cryo-EM density grid points, $N$ = number of atoms including water and ions, $W$ = 6 core workstation with one Intel E5-1650v4 @ 3.6 GHz CPU and one NVIDIA GTX 980 GPU, $S$ = 24 core server node with two Intel Gold 6146 @ 3.2 GHz CPUs and two GTX 1080Ti GPUs. $T_S$ = total run time for the short (40 ns) and long (70 ns) protocols in days using the $S$ hardware configuration (see **Figure 2—figure supplement 1**). The run times do not include setting up the simulated systems or batch queuing times.

*2-fs time step was used.

DOI: https://doi.org/10.7554/eLife.43542.028

## Code availability

For the results presented in this work, we used GROMACS version 5.0.7 to which our density fitting module was added. A tar archive of this version as well as the latest 2018 release with density fitting can be downloaded at https://www.mpibpc.mpg.de/grubmueller/densityfitting. For the benchmarks shown in *Table 9*, we used the latest 2018 version that we also recommend for general usage as it is faster by a factor of 1.3–1.8 (depending on the hardware used) as compared to the 5.0.7 version. We also provide installation instructions and a tutorial to set up and run the aldolase refinement simulation (*Figure 3*) which can be downloaded using the above link.

## Benchmarks

GROMACS 2018 was compiled with GCC 6.4, AVX2-256 SIMD instructions, CUDA 9.1, and OpenMP support using its built-in thread-MPI library. Calculations of the Coulomb and Van der Waals potentials and forces were offloaded to the GPU. Irrespective of the number of atoms in the simulated system, refinement performance was mainly dictated by the number of grid points in the density map, which explains the order of magnitude difference in timings between the tubulin (smallest) and the ribosome (largest) system. The highest performance was recorded on a single 24-core server node equipped with two consumer-class GPUs. In the non-ribosome cases, the complete refinement (half-map, full map, simulated annealing) took 0.4–2 days and 0.6–3.3 days on the faster hardware for the shortest and the longest protocols, respectively (*Figure 2—figure supplement 1*), respectively. The ribosome refinement took approximately 19 days using the shortest protocol. Currently, the density code used to calculate the correlation coefficient between the experimental and the simulated map poses a bottleneck. However, our preliminary code optimization for multiple GPU node usage showed almost a three fold performance improvement, bringing the refinement time for the ribosome system down to ~6 days when running on 8 GPU nodes. Moreover, we expect an additional gain in performance by a factor of 2 due to a less computationally expensive representation of the simulated map, yielding ~3.5 days for the ribosome refinement using the same number of GPU nodes.

## Structure availability

All refined structures presented in this work are provided as supplementary material.

## Acknowledgements

This work was supported by the Max Planck Society and the German Research Foundation via the research grant No. IG 109/1–1 (M Igaev) and FOR 1805 (AC Vaiana). The authors acknowledge computer time provided by the North-German Supercomputing Alliance (HLRN; located in Hannover and Berlin, Germany) as well as by the the Max Planck Computing and Data Facility and the Leibniz Supercomputing Centre (both located in Garching, Germany). The authors thank Holger Stark, Ashwin Chari, Niels Fischer and Wojciech Kopec (Max Planck Institute for Biophysical Chemistry, Göttingen, Germany), and Christian Blau (Stockholm University, Sweden) for insightful discussions. The authors also thank Mario Halic (LMU Munich, Germany), Rui Zhang (Washington University in St. Louis, USA) and Steven Wang (University of Illinois, Urbana-Champaign, USA) for kindly providing the nucleosome half-maps, the microtubule cryo-EM reconstructions, and the ReMDFF reference model of TRPV1, respectively.

## Additional information

### Funding

| Funder | Grant reference number | Author |
|---|---|---|
| Max-Planck-Gesellschaft | Open-access funding | Maxim Igaev<br>Carsten Kutzner<br>Lars V Bock<br>Andrea C Vaiana<br>Helmut Grubmüller |

| Deutsche Forschungsge-meinschaft | IG 109/1-1 | Maxim Igaev |
| Deutsche Forschungsge-meinschaft | FOR 1805 | Andrea C Vaiana |

The funders had no role in study design, data collection and interpretation, or the decision to submit the work for publication.

## Author contributions
Maxim Igaev, Conceptualization, Data curation, Formal analysis, Validation, Investigation, Visualization, Methodology, Writing—original draft, Project administration; Carsten Kutzner, Software, Methodology, Writing—review and editing; Lars V Bock, Data curation, Writing—review and editing; Andrea C Vaiana, Conceptualization, Data curation, Formal analysis, Investigation, Methodology, Writing—original draft, Project administration; Helmut Grubmüller, Conceptualization, Resources, Supervision, Methodology, Writing—review and editing

## Author ORCIDs
Maxim Igaev (iD) http://orcid.org/0000-0001-8781-1604
Andrea C Vaiana (iD) https://orcid.org/0000-0002-8865-0651
Helmut Grubmüller (iD) https://orcid.org/0000-0002-3270-3144

## Decision letter and Author response
Decision letter https://doi.org/10.7554/eLife.43542.038
Author response https://doi.org/10.7554/eLife.43542.039

## Additional files
### Supplementary files
• Transparent reporting form
DOI: https://doi.org/10.7554/eLife.43542.036

### Data availability
All structures generated or analyzed during this study are included in the supporting files. Refinement protocols and other methodologies are described in Materials and Methods.

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
