## [Decision Letter]

[Editors’ note: a previous version of this study was rejected after peer review, but the authors submitted for reconsideration. The first decision letter after peer review is shown below.]

Thank you for submitting your work entitled "Automated correlation-based structure refinement for high-resolution cryo-EM maps of large biomolecular complexes" for consideration by *eLife*. Your article has been reviewed by three peer reviewers, one of whom is a member of our Board of Reviewing Editors, and the evaluation has been overseen by Senior Editor. The following individual involved in review of your submission has agreed to reveal their identity: James S Fraser (Reviewer #3).

Our decision has been reached after consultation between the reviewers. Based on these discussions and the individual reviews below, we regret to inform you that your work will not be considered further for publication in *eLife*.

Grubmuller et al., use molecular dynamics refinement against a real-space map-correlation coefficient in combination with a chemically-accurate force field in the presence of explicit solvent. In order to better overcome local minima, the resolution of the model map was gradually increased starting at very low resolution to the maximum available resolution of the EM data. Moreover, high-temperature simulated annealing is used. Although each of these approaches individually are not new, this work describes a new implementation that combines all these approaches. However, there are many shortcomings that preclude publication of the present work in *eLife* as outlined in detail below. Briefly, the main concerns are the lack of comparison to other existing methods, unsubstantiated claims of model accuracy, the lack of other test cases (both at very high and medium-low resolution), and the potential of overfitting.

*Reviewer #1:*

Grubmuller et al., use molecular dynamics refinement against a real-space map-correlation coefficient in combination with a chemically-accurate force field in the presence of explicit solvent. In order to better overcome local minima, the resolution of the model map was gradually increased starting at very low resolution to the maximum available resolution of the EM data. Moreover, high-temperature simulated annealing is used. Although each of these approaches individually are not new, this work describes a new implementation that combines all these approaches. The method should be a useful new addition to the growing arsenal of tools for the refinement of three-dimensional models against maps obtained by single particle cryoEM.

Please comment on optimization of the resolution ramp, relative weight (force constant), and simulated annealing temperature vs. simulation time. The Materials and methods section suggests that somewhat different refinement protocols were used for each of the four example systems. What rationale was used to design these different protocols? More generally, please provide a guideline for selecting the various parameters.

There is a lack of a comparison to methods that are implemented in other programs, such as Phenix or Refmac. In particular, Phenix does have a simulated annealing option, although it does not use a full force field.

Afonine et al., 2018, showed that judicious use of restraints improves the quality of real-space refinement against medium/low cryo-EM density maps. The presence of such additional restraints may be beneficial during the low-resolution stages of the refinement. Please comment.

As an additional criterion for the performance of the refinements, please calculate the CaBLAM validation scores for the deposited structures and the re-refined structures. In particular, the CaBLAM scores (Richardson, J. S., et al. Acta Crystallographica Section D74, 132-142. (2018), doi:10.1107/S2059798317009834; implemented in Phenix) asses the geometry of peptide bonds and the geometry of Ca atoms.

*Reviewer #2:*

This well-written manuscript describes methods for the refinement of atomic models against cryo-EM reconstructions. The basis of the method is the inclusion of a biasing term in a molecular dynamics potential that increases the overlap of the model with the experimental map. This approach is not new, and has been used in programs such as MDFF. The authors do describe a new approach to gradually increasing the resolution of the data in refinement, and the application of simulated annealing protocols. However, the limited novelty of the work, and other considerations significantly reduce my enthusiasm. Some important points for consideration:

In the Abstract the authors make several claims for the work:

"The method utilizes a chemically accurate force field and exhaustive thermodynamic sampling while efficiently improving the cross-correlation between the modeled structure and the cryo-EM map. For several test systems of different size with resolutions 2.5-3.4 Å, it generates stereochemically accurate models from cryo-EM data with a low risk of overfitting. The presented automated framework allows efficient refinement of large molecular complexes such as the ribosome without manual intervention or any additional ad hoc restraints, providing a good balance between stereochemical accuracy, refinement time, and hardware requirements."

There are several problems:

- While it is true that the models are improved in some ways after the refinement protocol, in general the rotamer conformations are significantly poorer (outside the realm of what is acceptable for a protein structure), and the Ramachandran distributions also poorer. Therefore the claim of "stereochemically accurate models" is overreaching.

- "Efficient refinement" clearly means different things to different people. For the systems studied the methods described took between 2 and 25 days running on a multiple core system with 2 GPUs. For many researchers this would be considered very long run times, especially when there are other methods that produce comparable results in much shorter times.

- A minor quibble, but the phrase "exhaustive thermodynamic sampling", suggests just that. However, the protocols described are unlikely to be exhaustive. They employ simulated annealing methods which do expand the conformational space that is sampled, but not exhaustively.

There are other issues that need to be considered:

- The authors apply their method to 4 test cases, including a large ribosome structure. However, this is a limited sampling of models, and in particular doesn't include models with very poor starting geometries. All 4 test cases are of high quality to begin with – as judged by the EMRinger score, Molprobity score and other presented criteria. The manuscript would be much more persuasive if it presented results for a larger number of structures – it should be noted that other researchers have applied automated refinement methods to all suitable cryo-EM structures available from the Protein Data Bank. In particular it would be helpful to see results for models with significant initial structural errors.

- The work, as presented, focuses on higher resolution cryo-EM data (3Å or higher is stated in the Abstract). There is definitely a need for accurate refinement methods at these resolutions. However, the quality of starting models at these resolutions is typically very high, and often don't present significant challenges for refinement algorithms. In addition, it is worth noting that, as of a few weeks ago, there are 107 structures from 3Å or better cryo-EM data, 808 structures between 3Å and 4Å resolution, and 515 structures between 4Å and 5Å resolution. The 3.5Å to 5Å resolution range often presents some serious challenges for both model building and subsequent refinement. The application of the methods to 3.1Å and 3.5Å test cases in the manuscript is applauded. However, the manuscript would have been more exciting if it presented a more thorough analysis of the application of the method at resolutions worse that 3.5Å.

- The authors could consider test cases where there is some higher resolution "gold standard" available to validate the results of refinement with lower resolution data. This is, of course, non-trivial, but very helpful in confirming the accuracy and radius of convergence of the refinement method.

- As noted above, much of what is described has been introduced in earlier works, so the novelty is limited. The approach to a gradual increase in data resolution is interesting – blurring of the model density only. However, the authors don't demonstrate that this is an improvement over previous methods.

- Simulated annealing has been already applied in other programs for model to map fitting (DEN, MDFF). This is also used, as an option, in the phenix.real_space_refine program.

- Other approaches to model refinement have been implemented, with the goal of stereochemical quality and computational efficiency. In particular the phenix.real_space_refine program is not mentioned in the manuscript. This program has been used for the successful refinement of many cryo-EM models.

- In the description of the proteasome 3.1A refinement it is stated that "The improved quality statistics for our model (Figure 4A, top-right) now indicate that the conformations of these loops were refined more accurately". How do the authors know this? The model may have improved overall, while the confirmation of these particular loops may be worse after refinement.

*Reviewer #3:*

The major goal of this paper is to use molecule dynamics simulations to improve the fit and geometry of structures refined against high resolution (better than 4A) maps from single particle cryoEM. The principle contribution is to combine advances (classic work on simulated annealing, MDFF, and recent work from the Tama group) with improved MD and map calculation codes. The manuscript is beautifully formatted and the figures are very clear. The software is benchmarked against a very limited set of targets, focusing on experimental data and previously known crystal structures, which mimics its potential application in the real world. The ribosome dataset seems to be the only one somewhat challenging case, but even that consists largely of a rigid body rotation with few internal degrees of freedom changing during the low resolution part of the search. Given the long run time of the method, and the lack of comparisons against existing software (REFMAC, MDFF, phenix.real_space_refine), it is unclear how widely used this package will become. The major weaknesses of the manuscript are lack of benchmarking against these methods and whether the radius of convergence will hold for poorer starting models (e.g. structures based on homology modeling, etc.).

[Editors’ note: what now follows is the decision letter after the authors submitted for further consideration.]

Thank you for submitting your article "Automated cryo-EM structure refinement using correlation-driven molecular dynamics" for consideration by *eLife*. Your article has been reviewed by three peer reviewers, including Axel Brunger as the Reviewing Editor, and the evaluation has been overseen by John Kuriyan as the Senior Editor. The following individual involved in review of your submission has agreed to reveal their identity: James S Fraser (Reviewer #2).

The Reviewing Editor and the reviewers believe that this paper would be of considerable general interest, provided that the validity of the method can be demonstrated in a convincing way. They continue to feel, however, that the demonstration that the method performs better than alternative methods is still not convincing. After discussion, we are prepared to consider a revised manuscript that addresses the concerns raised by reviewers. The eventual acceptance of this paper hinges on a satisfactory resolution of these issues. We recognize that the revisions would require a substantial amount of new work. If this is not possible, you might choose to submit this work elsewhere in its present form.

Summary:

In this revised manuscript, the authors have addressed many, but not all, of the concerns that were raised. The use of a "complete" empirical force field (rather than the simplified force fields commonly used in macromolecular refinement packages) seems to contribute to the differences when compared with other methods, along with the use of simulated annealing, and ramping up the resolution. While each of these individual approaches is not new, the authors suggest that the combination of them produces better quality EM structures. If this claim can be substantiated, it would represent a welcome new tool in the arsenal of refinement tools for EM structures. However, we are not convinced that the current manuscript definitely demonstrates such superiority, and major revisions are required for further consideration of this manuscript as outlined in the following.

Critical points to address:

1) Overall, while the manuscript has been improved by inclusion of multiple test-cases of different quality maps and comparisons to alternative (standard) methods. However, distinct alternative approaches were used for each test case and not applied across all examples. For example, MDFF was initially developed primarily for lower (~5-10 A) resolution cases, but is not used for the CorA example here (perhaps the strongest new result in the paper). Please consistently use a common set of alternative (standard) methods for all cases, and, additionally, use methods such as MDFF for the low resolution cases.

2) NSF test case, subsection “N-ethylmaleimide sensitive factor (NSF): comparison with Phenix”: for the re-refinement of the recent NSF/SNAP/SNARE EM single particle structure by White et al., 2018 (Figure 7 in the paper), the most significant advantage appears to be the substantially lower clash score compared to phenix refinement. The authors seem to be able to do better (especially for the clash score), without extensive grid searching (at least as far as we can tell). However, the clashscore itself does not indicate if their refined models are closer to the true structure. We had previously asked for the CABLAM analysis that should be included. Nevertheless, ultimately, the question arises if the re-refined structure is closer to the true structure of the 20S complex. Since there is no higher resolution structure of the entire complex available, the authors could test the quality of their refinement with the D2 domain of NSF by comparing it with the available high-resolution crystal structures.

Other important points to address:

1) NSF test case: "Finally we again point out the strong anti-correlation between the number of steric clashes and rotamer/Ramachandran outliers in the deposited reference and our models (discussed later in the Results section).” Which anti-correlation does this statement refer to?

2) Figure 7: the deposited 6MDO structure (referred to as the "reference" structure) actually has rather reasonable geometry. The improvement by the author's method is primarily a reduction of the clash score (Figure 7B). Please comment and speculate about the reasons (e.g., use of an accurate force field might clean up poorly determined regions?). Note that the Ramachandran statistics also slightly improves, contrary to the suggested anti-correlation (see previous point).

3) Figure 7A: Please calculate the heat map but compared to 6MDO instead of 3J94 as well since 6MDO and 3J94 are not equivalent models (e.g., they just pasted the SNAP-25 N terminal residues into 3J94, in which those residues do not exist in 3J94).

4) Figure 7A: it appears that their method mostly fixed the geometry of the solvent exposed parts of the model, so likely the ATP binding pocket is pretty much the same? These differences could possibly also arise from changes in rigid body positions of the domains in the hexamer. Please comment.

5) Figure 7: clarify how the various parameters for their refinement were chosen. Was there a lot of trial and error involved?

6) "Furthermore, the NSF refinement demonstrates the usefulness of the half-map-based cross-validation procedure as, for a refinement against a full map only, much more refinement runs have to done to identify the optimal map bias". Where is it is shown that more refinement runs have to be performed without access to half maps?

7) The question of whether a single model or an ensemble is more appropriate is worth discussing. For example, the three TRPV1 structures produced by independent refinement vary across the model (Figure 5—figure supplement 2), and may suggest that an ensemble refinement may be suitable. Moreover, the discussion of "ideal geometry and over/underfitting" in the last paragraph of the subsection “Force-field dependence of the model geometry” completely ignores the effect of conformational heterogeneity that may be related to the non-ideal nature of refining a single conformation against an EM map. Ensemble averaging misinterpreted as a single structure is a possible reason. See Rice, Shamoo and Brünger, 1998, and Burnley et al., 2012 for crystallographic multi-start and ensemble refinements, respectively, and Herzik, Fraser and Lander et al., 2018 and Bonomi, Pellarin and Vendruscolo, et al.2018 for EM structure ensemble refinement. On the other hand, low determinacy of 3D classification and refinement of the particles may be another reason for the non-ideal nature of the refinement. These two issues are likely convoluted and represent a major challenge for the interpretation of single particle EM data. The authors are not expected to perform ensemble refinements, but rather, a discussion of this point is requested.

8) Note that the use of an empirical force field for crystallographic refinement was first suggested by Brunger et al., 1987, and re-investigated in more recent papers, e.g., Fenn et al., 2011. A discussion of this point would be appropriate.

9) How were the MolProbity statistics calculated? There are some subtleties that might need to be taken into consideration. By default Molprobity uses bond lengths for hydrogen atoms that are centered on the X-ray scattering position (electron cloud), as opposed to the nuclear position of the atom – i.e. the former is shorter than the latter (see Protein Sci. 2018, 27:293-315). This can have some consequences depending on the workflow. If MolProbity is given a model containing hydrogen atoms whose positions have been defined using nuclear positions the clash score may be reported as high. Alternatively, if a model is optimized using nuclear position hydrogen atoms, and these are removed, MolProbity automatically adds hydrogen atoms at the electron cloud position, resulting in systematically low clash scores. The authors should consider their protocol in light of this, and make it clear what was done in the Materials and methods section.

10) Is a complete model a requirement for the method? In the Materials and methods section it appears that any missing residues in the test structures were "filled in" using homology modeling or other approaches. Is this mandatory? If so it needs to be clearly communicated in the paper. The creation of parts of models for which there is no clear experimental data to support that model are questionable.

11) The authors make an interesting argument about clashes and side- and mainchain outliers. However, arguing that a very low clash score and a significant number of geometry outliers is reasonable stretches credulity. It is well known from high resolution structures what the expectations are for outliers. It is also observed that at lower resolution it becomes increasingly hard to determine correct rotamer and mainchain conformations based on the experimental data. The authors might be better off to emphasize that they are able to create models with a quality consistent with many other methods at this resolution, often better than the starting models, and accept that there is room for improvement.

12) The authors should be clear what they mean when they use the phrase "overfitting" in this context. There is often confusion about this term's meaning – e.g. fitting of noise, model correctness, determination of the optimal protocol.

13) It would seem appropriate to mention the work of Sanbonmatsu and colleagues: Kirmizialtin et al., 2015.

[Editors' note: further revisions were requested prior to acceptance, as described below.]

Thank you for resubmitting your work entitled "Automated cryo-EM structure refinement using correlation-driven molecular dynamics" for further consideration at *eLife*. Your revised article has been favorably evaluated by John Kuriyan (Senior Editor), a Reviewing Editor, and three reviewers.

We thank you for addressing many of our concerns, in particular with the new discussion sections, the CorA analysis, and the comparisons for all refinements. The application to NSF shows that the method can obtain a model that is closer to the true structure as assessed by the agreement with the high-resolution crystal structures of the NSF D2 domain. In particular, the improved placement of the nucleotide is impressive. However, there are some remaining minor issues that need to be addressed before acceptance, as outlined below:

1. Please provide the CABLAM and other statistics (provided in the supplemental files as. tex files) as proper tables.

2. You are correct in their assertion that the CDMD procedure has a larger radius of convergence than the Refmac, Phenix and Rosetta procedures. However, while it is very much appreciated that making use of other programs can be challenging, it is probable that some of the protocols used are not optimal for the particular cases being studied. For example, in the case of phenix.real_space_refine the default protocols are appropriate for refinement of models close to correct, i.e. not significantly displaced from the cryo-EM volume (globally or locally). Moreover, the protocol outlined in White et al., 2018 helps extend the radius of convergence. However, for models that have more significant starting errors, morphing is an important option for increasing the radius of convergence. It can also be necessary to run more refinement cycles to increase the convergence radius. Please comment in the Discussion.

3. The benchmark information in Table 2 is likely to be confusing for many readers (the timings provided in the Benchmark text are very useful). Would it be possible to provide statistics in the form of total run times for refinements in a table form?

---

## [Author Response]

[Editors’ note: the author responses to the first round of peer review follow.]

Many thanks for handling our manuscript and for the constructive feedback. It has helped us to completely rewrite and re-structure our manuscript. As you will see from our detailed replies to the reviewers’ comments below, we have now addressed all major issues, with a particular emphasis on the two main ones: (1) As requested, we now provide a thorough assessment of our method using a much larger and more diverse set of test cases, covering the most typical refinement challenges posed by modern cryo-EM: poor starting models, low- and medium-resolution maps, highly heterogeneous local map resolutions, or combinations thereof. (2) Concerning the lack of comparison to other methods in the original manuscript, we now carefully analyze and discuss in detail the differences between our results and previously published structures refined with other widely used methods, in terms of both model accuracy and potential of overfitting. For some test cases, we now present a direct comparison to the most recent versions of the well-established methods (Rosetta, ReMDFF and Phenix). Overall, we are convinced that the manuscript now meets the high standards of *eLife* and provides a substantial contribution to the field.

Reviewer #1:

[…] Please comment on optimization of the resolution ramp, relative weight (force constant), and simulated annealing temperature vs. simulation time. The Materials and methods section suggests that somewhat different refinement protocols were used for each of the four example systems. What rationale was used to design these different protocols? More generally, please provide a guideline for selecting the various parameters.

Indeed, the rationale behind the protocols and the choice of the refinement parameters (resolution and force constant ramps, simulated annealing temperature etc.) was not sufficiently explained in the original manuscript. We have now added this information as well as a schematic representation of the refinement and validation protocols in a supplementary figure (Figure 2—figure supplement 1). The Materials and methods section now also contains guidelines for selecting the various parameters and protocol durations. The tutorial containing instructions and files necessary to set up the aldolase refinement (Figure 3) additionally explains technical aspects.

There is a lack of a comparison to methods that are implemented in other programs, such as Phenix or Refmac. In particular, Phenix does have a simulated annealing option, although it does not use a full force field.

The manuscript has been completely revised to include (a) more test systems to challenge the method in different, orthogonal ways (Figures 3-10), (b) a more detailed analysis and discussion of the obtained models in the context of the previously published structures, and (c) direct comparisons to other widely used refinement methods (see Figure 5 for a COOT comparison, Figure 6A, B for a Rosetta comparison, Figure 6C, D for a ReMDFF comparison, and Figure 7 for a Phenix comparison). More specific changes are described further below.

Afonine et al., 2018, showed that judicious use of restraints improves the quality of real-space refinement against medium/low cryo-EM density maps. The presence of such additional restraints may be beneficial during the low-resolution stages of the refinement. Please comment.

While it is true that rotamer, Ramachandran and secondary structure restraints used in some refinement algorithms have been shown to improve the quality of fitted structures in some cases, we prefer not to mix force field parameters and knowledge-based restraints. We think that modern atomistic force fields should already contain all necessary information to maintain accurate model geometries and, therefore, that the addition of knowledge-based restraints may create conflicting optimization goals. Therefore, our approach, by construction, does not contain such restraints. As demonstrated by the structure quality assessment for all of the refined systems, our approach also does not require those. The use of the “soft” (global) correlation biasing potential further reduces the risk of structural deteriorations.

Of course, our implementation technically allows one to introduce such restraints, which are already part of the GROMACS functionality.

Reviewer #2:

[…] In the Abstract the authors make several claims for the work:"The method utilizes a chemically accurate force field and exhaustive thermodynamic sampling while efficiently improving the cross-correlation between the modeled structure and the cryo-EM map. For several test systems of different size with resolutions 2.5-3.4 Å, it generates stereochemically accurate models from cryo-EM data with a low risk of overfitting. The presented automated framework allows efficient refinement of large molecular complexes such as the ribosome without manual intervention or any additional ad hoc restraints, providing a good balance between stereochemical accuracy, refinement time, and hardware requirements."There are several problems:- While it is true that the models are improved in some ways after the refinement protocol, in general the rotamer conformations are significantly poorer (outside the realm of what is acceptable for a protein structure), and the Ramachandran distributions also poorer. Therefore the claim of "stereochemically accurate models" is overreaching.

It is true that, for some of our models, the Ramachandran distributions were poorer after refinement as compared to the deposited reference structures. It would be nevertheless highly surprising if it were not the case. While the deposited structures (except for the ReMDFF model) were built with structural restraints enforced and, therefore, by construction, did not violate the “ideal” rotamer and Ramachandran distributions, this came at the cost of significant steric clashes (in many cases tens per thousand atoms or more). In contrast, the force field in our approach avoided steric clashes at the cost of, in our view, tolerable rotamer and Ramachandran outliers. Furthermore, the above situation is far from being the general rule – in fact, we saw improvements in rotamer geometries for the TRPV1 (Figure 5, Figure 6C, D) and nucleosome (Figure 8) systems which came together with the drastic reduction in steric clashes, and the rotamer quality of the tubulin system (Figure 4) was comparable with that of the higher-resolution control structures. The TRPV1 (Figure 6C, D) and ribosome (Figure 9) cases, in turn, demonstrated that increased Ramachandran outliers are not a strict trend and can be reduced by our method.

We have now added a new section at the end of the Results, where we specifically summarize – using now a much larger set of test cases – the structure quality statistics aggregated from all the test cases and thoroughly discuss the “acceptability” of rotamer and Ramachandran outliers in the general context of structure refinement.

- "Efficient refinement" clearly means different things to different people. For the systems studied the methods described took between 2 and 25 days running on a multiple core system with 2 GPUs. For many researchers this would be considered very long run times, especially when there are other methods that produce comparable results in much shorter times.- Other approaches to model refinement have been implemented, with the goal of stereochemical quality and computational efficiency. In particular the phenix.real_space_refine program is not mentioned in the manuscript. This program has been used for the successful refinement of many cryo-EM models.

The reviewer notes that the computational efficiency of our refinement method is below that of several other methods. Although these methods do achieve shorter run times, we respectfully disagree with the statement that the obtained results are comparable. Whereas these methods perform similarly well for maps at near-atomic resolution (Figure 3 and Figure 6A, B), the additional hard test cases we now show that our method performs better, in particular at low and spatially heterogeneous local resolutions. Concerning the run times, for small to medium-sized systems (Figure 3-8 and Figure 10), these range from 0.5 to 3 days, which we would consider acceptable given the higher quality of the resulting models and the method’s universality. For large systems (Figure 9), the refinement performance is mainly dictated by the large number of grid points in the density map, which is indeed the main bottleneck in the current implementation of the density calculation code. Preliminary code optimization for multiple GPU node usage showed almost a 3-fold performance improvement, thus bringing the refinement time for the ribosome system down to 6-7 days when running on 8 GPU-nodes. Moreover, we expect an additional gain in performance by a factor of 2 due to a less computationally expensive representation of the simulated map, eventually yielding ~3.5 days for the ribosome refinement using the same number of GPU nodes. For a fully automated method, we think this is acceptable.

- A minor quibble, but the phrase "exhaustive thermodynamic sampling", suggests just that. However, the protocols described are unlikely to be exhaustive. They employ simulated annealing methods which do expand the conformational space that is sampled, but not exhaustively.

We agree with the reviewer and have removed this phrasing in the revised manuscript.

There are other issues that need to be considered:- The authors apply their method to 4 test cases, including a large ribosome structure. However, this is a limited sampling of models, and in particular doesn't include models with very poor starting geometries. All 4 test cases are of high quality to begin with – as judged by the EMRinger score, Molprobity score and other presented criteria. The manuscript would be much more persuasive if it presented results for a larger number of structures – it should be noted that other researchers have applied automated refinement methods to all suitable cryo-EM structures available from the Protein Data Bank. In particular it would be helpful to see results for models with significant initial structural errors.

We agree with the reviewer that the previous set of test cases was limited and did not entirely reflect the full spectrum of refinement challenges posed by modern cryo-EM. In the revised manuscript, following the reviewer’s advice, we have substantially extended the set of refinement cases to include more challenging systems. We particularly added cases for which the starting structures had significant initial errors.

- The work, as presented, focuses on higher resolution cryo-EM data (3Å or higher is stated in the Abstract). There is definitely a need for accurate refinement methods at these resolutions. However, the quality of starting models at these resolutions is typically very high, and often don't present significant challenges for refinement algorithms. In addition, it is worth noting that, as of a few weeks ago, there are 107 structures from 3Å or better cryo-EM data, 808 structures between 3Å and 4Å resolution, and 515 structures between 4Å and 5Å resolution. The 3.5Å to 5Å resolution range often presents some serious challenges for both model building and subsequent refinement. The application of the methods to 3.1Å and 3.5Å test cases in the manuscript is applauded. However, the manuscript would have been more exciting if it presented a more thorough analysis of the application of the method at resolutions worse that 3.5Å.

Again, we agree that considering only high-resolution cases (~3.5 Å or higher) was insufficient to assess the method’s convergence and accuracy. In the new, extended set of test cases, a much wider range of resolutions (2.6-7.1 Å) is covered.

- The authors could consider test cases where there is some higher resolution "gold standard" available to validate the results of refinement with lower resolution data. This is, of course, non-trivial, but very helpful in confirming the accuracy and radius of convergence of the refinement method.

We thank the reviewer for this suggestion and have now included such a test case (Figure 4).

- As noted above, much of what is described has been introduced in earlier works, so the novelty is limited. The approach to a gradual increase in data resolution is interesting – blurring of the model density only. However, the authors don't demonstrate that this is an improvement over previous methods.- Simulated annealing has been already applied in other programs for model to map fitting (DEN, MDFF). This is also used, as an option, in the phenix.real_space_refine program.- Other approaches to model refinement have been implemented, with the goal of stereochemical quality and computational efficiency. In particular the phenix.real_space_refine program is not mentioned in the manuscript. This program has been used for the successful refinement of many cryoEM models.

Although many individual components of the proposed refinement method are, indeed, not new (e.g., correlation biasing potential or simulated annealing), the actual novelty is the integration of these components into an automated protocol involving a chemically accurate force field, the "soft" (nonlocal) correlation biasing potential coupled to adaptive resolution, and the half-map-based validation scheme, and we demonstrate, using the extended set of refinement cases, that this is indeed an improvement over previous methods. Unlike other methods, this integration allows fully automated refinement into density maps with heterogeneous local resolutions, as typical of modern cryo-EM maps. We now also explicitly compare our results to the deposited reference models and discuss in the context of other refinement methods used, including the most recent version of phenix.real_space_refine (Afonine et al., 2018) with optimized target functions (White et al., 2018).

- In the description of the proteasome 3.1A refinement it is stated that "The improved quality statistics for our model (Figure 4A, top-right) now indicate that the conformations of these loops were refined more accurately". How do the authors know this? The model may have improved overall, while the confirmation of these particular loops may be worse after refinement.

We have now removed the proteasome refinement case from the revised manuscript, as it was very similar to the aldolase refinement (Figure 3) both in terms of map resolution and starting structure quality.

Reviewer #3:

The major goal of this paper is to use molecule dynamics simulations to improve the fit and geometry of structures refined against high resolution (better than 4A) maps from single particle cryoEM. The principle contribution is to combine advances (classic work on simulated annealing, MDFF, and recent work from the Tama group) with improved MD and map calculation codes. The manuscript is beautifully formatted and the figures are very clear. The software is benchmarked against a very limited set of targets, focusing on experimental data and previously known crystal structures, which mimics its potential application in the real world. The ribosome dataset seems to be the only one somewhat challenging case, but even that consists largely of a rigid body rotation with few internal degrees of freedom changing during the low resolution part of the search. Given the long run time of the method, and the lack of comparisons against existing software (REFMAC, MDFF, phenix.real_space_refine), it is unclear how widely used this package will become. The major weaknesses of the manuscript are lack of benchmarking against these methods and whether the radius of convergence will hold for poorer starting models (e.g. structures based on homology modeling, etc.).

We believe that all of the concerns raised by reviewer #3 have now been addressed, as described above. The fact that there was such a big overlap between these concerns and those of the other reviewers brought us to pay particular attention to (a) substantially expanding the set of challenging test cases and (b) comparing against existing methods (Phenix, Rosetta, MDFF, Refmac, COOT).

[Editors' note: the author responses to the re-review follow.]

Critical points to address:1) Overall, while the manuscript has been improved by inclusion of multiple test-cases of different quality maps and comparisons to alternative (standard) methods. However, distinct alternative approaches were used for each test case and not applied across all examples. For example, MDFF was initially developed primarily for lower (~5-10 A) resolution cases, but is not used for the CorA example here (perhaps the strongest new result in the paper). Please consistently use a common set of alternative (standard) methods for all cases, and, additionally, use methods such as MDFF for the low resolution cases.

We are glad the reviewers agree that the inclusion of more test cases has improved the original version of our manuscript. Further encouraged by the reviewers, we have added a full, “all-structures-by-all-methods” comparison using the same far-away starting structures across all test cases. Considering that many of the tested methods have *not* been originally optimized to work with distant starting models or DNA/RNA, we think that, taken together, these comparisons strengthen our main conclusions. These new comparisons indeed demonstrate that the correlation-driven MD refinement has a much larger radius of convergence than current non-MD methods. As requested, we also provide a comparative refinement of the CorA system (Figure 10) using MDFF, which has helped to gain a clearer picture.

The manuscript has been modified as follows:

- Phenix, Rosetta and Refmac have been chosen as alternative methods for comparison for the test cases in Figures 3-9. The radius-of-convergence analysis is shown in the supplementary figures corresponding to Figures 3-9. The results of the comparative refinements are additionally discussed as separate paragraphs in each of the Results subsections. A separate subsection in Materials and methods is dedicated to the description of the Phenix/Rosetta/Refmac refinement protocols.

- MDFF has been applied to the CorA system (Figure 10), and the corresponding refinement protocol is described in Materials and methods. For consistency with MDFF, we have re-refined the CorA starting structure with our method but using a different force field (CHARMM36m), as CHARMM22* used in Figures 3-9 has no version compatible with NAMD. This has also allowed a fairer comparison of our method with MDFF.

2) NSF test case, subsection “N-ethylmaleimide sensitive factor (NSF): comparison with Phenix”: for the re-refinement of the recent NSF/SNAP/SNARE EM single particle structure by White et al., 2018 (Figure 7 in the paper), the most significant advantage appears to be the substantially lower clash score compared to phenix refinement. The authors seem to be able to do better (especially for the clash score), without extensive grid searching (at least as far as we can tell). However, the clashscore itself does not indicate if their refined models are closer to the true structure. We had previously asked for the CABLAM analysis that should be included. Nevertheless, ultimately, the question arises if the re-refined structure is closer to the true structure of the 20S complex. Since there is no higher resolution structure of the entire complex available, the authors could test the quality of their refinement with the D2 domain of NSF by comparing it with the available high-resolution crystal structures.

We agree with the reviewers that an improvement (however pronounced it is) in any of the model quality metrics relatively to the published reference model does not per se indicate that the resulting model is closer to the “true structure”, and that a higher-quality control structure would be required for validation (ideally derived from a higher-resolution data set, e.g., X-ray). We therefore welcomed the suggestion by the reviewers and have validated our refined models in those test cases where higher resolution structures of either the entire refined system or parts of it were available (including NSF). The only exception was TRPV1 for which we could not find a high-resolution crystal structure of the channel or close homolog structures. The results of the validations are shown in panels D of Figures 39 and are now discussed in separate paragraphs in each of the Results subsections. Also following the request to calculate CaBLAM scores, all quality metrics including CaBLAM are now summarized in separate supplementary tables to each main text figure.

Regarding the grid searching mentioned by the reviewers, indeed, we did not perform any extensive parameter search as in the original study by White et al. The major challenge in this particular refinement case was the absence of the half-maps which could have drastically simplified the choice of the optimal force constant. As the half-maps were not available, we did a scan of the force constant space by performing seven independent refinements (instead of just one like in, e.g., Figure 3) with different target force constants (Figure 7B). The optimal force constant for the NSF system turned out to be close to the initial guess – 4-5 × 10^5^ kJ / mol – which was comparable with the one we used to refine the TRPV1 system. Other than that, there was no trial and error involved. This has now been explained more clearly in the main text.

Other important points to address:1) NSF test case: "Finally we again point out the strong anti-correlation between the number of steric clashes and rotamer/Ramachandran outliers in the deposited reference and our models (discussed later in the Results section).” Which anti-correlation does this statement refer to?

This statement refers to the reduction of steric clashes by our method that often comes at the price of increased rotamer and/or Ramachandran outliers, if one compares the refined models with the deposited ones. Examples are, *e.g.*, Figure 3b (bottom, gray vs. green), Figure 4B (bottom, gray and dark gray vs. cyan), Figure 6b (bottom, gray vs. pink, rotamers), Figure 9B (bottom, gray vs. orange, rotamers). We have now included a more detailed discussion of this observation in the last subsection of the Results (“Force field influence and model accuracy”). Additionally, we now avoid the repetitive use of this statement in the main text, mentioning it only once when describing the results of the aldolase refinement.

2) Figure 7: the deposited 6MDO structure (referred to as the "reference" structure) actually has rather reasonable geometry. The improvement by the author's method is primarily a reduction of the clash score (Figure 7B). Please comment and speculate about the reasons (e.g., use of an accurate force field might clean up poorly determined regions?). Note that the Ramachandran statistics also slightly improves, contrary to the suggested anti-correlation (see previous point).

As requested, we have added a paragraph to the section dedicated to the NSF refinement (Figure 7) discussing the difference between the reference structure (6MDO) and our model in more detail. See also our response in point (4) below.

3) Figure 7A: Please calculate the heat map but compared to 6MDO instead of 3J94 as well since 6MDO and 3J94 are not equivalent models (e.g., they just pasted the SNAP-25 N terminal residues into 3J94, in which those residues do not exist in 3J94).

For each refinement case (Figures 3-10), panel (A) in the corresponding figures shows the per-residue (or per-nucleotide) RMSD between the distant starting structure and the deposited reference. The same applies to Figure 7A where the heat map shows the per-residue RMSD between 6MDO and the distant starting structure. This was not sufficiently clearly stated in the previous manuscript version and is now explicitly pointed out in the respective figure legends.

4) Figure 7A: it appears that their method mostly fixed the geometry of the solvent exposed parts of the model, so likely the ATP binding pocket is pretty much the same? These differences could possibly also arise from changes in rigid body positions of the domains in the hexamer. Please comment.

While it is true that the solvent exposed parts of the NSF model were improved in our model, this was not the only difference between our model and 6MDO. In fact, the validation using higher-resolution X-ray structures (referred to as “controls” in our work), as suggested by the reviewers in Critical point (2), has helped identify the ATP binding pocket as yet another region in the NSF protomers that also differs considerably between the two models. This validation has revealed that either the adenosine group of ATP is rotated by 90° or 180° (3 of 6 NSF protomers in 6MDO) or the phosphate tail of ATP deviates considerably from that of the controls (all protomers in 6MDO). In contrast, our model reproduced the structure of the ATP binding pocket equally well across the entire D2 hexamer. As mentioned above, we have added a paragraph to the NSF section discussing these aspects in more detail.

With respect to global rigid-body positioning of the domains in the hexamer, we have not found any difference between 6MDO and our model, which is also supported by the very similar reciprocal correlation at low spatial frequencies (Figure 7B, top).

5) Figure 7: clarify how the various parameters for their refinement were chosen. Was there a lot of trial and error involved?

See our response to Critical point (2). The improved versions of the sections “Simulation setup and refinement” and “Cross-validation and stereochemical quality” in Materials and methods in combination with the protocol summary in Figure 2—figure supplement 1 now provide a guide for how the refinement parameters were chosen. In particular, we have added a paragraph to the section “Cross-validation and stereochemical quality” that specifically elucidates how to proceed in refinement situations in which the half-maps are not available.

In some cases, for example, several half-map refinements at different temperatures (see Figure 2—figure supplement 1) were performed to test the method’s robustness. In other cases (e.g., the nucleosome refinement), independent refinements with different force constants were done to test the underfitting hypothesis. Overall, very little trial and error was required – the fact that the whole refinement procedure is fully automated really paid off.

6) "Furthermore, the NSF refinement demonstrates the usefulness of the half-map-based cross-validation procedure as, for a refinement against a full map only, much more refinement runs have to done to identify the optimal map bias". Where is it is shown that more refinement runs have to be performed without access to half maps?

See our response to point (5). We have now clarified this issue in the revised manuscript by explicitly stating why and how the absence of half-maps affected the NSF refinement in the first paragraph of the corresponding section of the main text.

7) The question of whether a single model or an ensemble is more appropriate is worth discussing. For example, the three TRPV1 structures produced by independent refinement vary across the model (Figure 5—figure supplement 2), and may suggest that an ensemble refinement may be suitable. Moreover, the discussion of "ideal geometry and over/underfitting" in the last paragraph of the subsection “Force-field dependence of the model geometry” completely ignores the effect of conformational heterogeneity that may be related to the non-ideal nature of refining a single conformation against an EM map. Ensemble averaging misinterpreted as a single structure is a possible reason. See Rice, Shamoo and Brünger, 1998, and Burnley et al., 2012 for crystallographic multi-start and ensemble refinements, respectively, and Herzik, Fraser and Lander, 2018 and Bonomi, Pellarin and Vendruscolo, 2018 for EM structure ensemble refinement. On the other hand, low determinacy of 3D classification and refinement of the particles may be another reason for the non-ideal nature of the refinement. These two issues are likely convoluted and represent a major challenge for the interpretation of single particle EM data. The authors are not expected to perform ensemble refinements, but rather, a discussion of this point is requested.

We agree and thank the reviewers for pointing out these important issues and now discuss them in more detail in the last two paragraphs of the section “Force field influence and model accuracy”. Briefly, we now explicitly distinguish between the two sources of errors in our refined models: (a) conformational heterogeneity and poor 3D classification convoluted in cryo-EM data and (b) the imperfect nature of atomistic force fields. Each point is discussed in a separate paragraph. Besides citing the above literature on ensemble refinement, we also explicitly state that the present work is limited to single-structure refinements.

8) Note that the use of an empirical force field for crystallographic refinement was first suggested by Brunger et al., 1987, and re-investigated in more recent papers, e.g., Fenn et al., 2011. A discussion of this point would be appropriate.

We thank the reviewers for pointing this out. The mentioned studies as well as a comprehensive review on the use of molecular dynamics in cryo-EM refinement by Sanbonmatsu and coworkers are now cited and acknowledged.

9) How were the MolProbity statistics calculated? There are some subtleties that might need to be taken into consideration. By default Molprobity uses bond lengths for hydrogen atoms that are centered on the X-ray scattering position (electron cloud), as opposed to the nuclear position of the atom – i.e. the former is shorter than the latter (see Protein Sci. 2018, 27:293-315). This can have some consequences depending on the workflow. If MolProbity is given a model containing hydrogen atoms whose positions have been defined using nuclear positions the clash score may be reported as high. Alternatively, if a model is optimized using nuclear position hydrogen atoms, and these are removed, MolProbity automatically adds hydrogen atoms at the electron cloud position, resulting in systematically low clash scores. The authors should consider their protocol in light of this, and make it clear what was done in the Materials and methods section.

This information, indeed, was not provided in the previous manuscript version. Throughout the study, we consistently used X-ray scattering positions (electron cloud, default option) for all structures. When using the nuclear position option in MolProbity, we noticed that the number of steric clashes increased by 15-20%. This, however, had almost no effect on the MolProbity scores of the structures refined with our method or MDFF (CorA in Figure 10), as those had almost zero steric clashes by construction, independently of the hydrogen placement option used. We have now made this clear and have added all required details in the section “Cross-validation and stereochemical quality” in Materials and methods.

10) Is a complete model a requirement for the method? In the Materials and methods section it appears that any missing residues in the test structures were "filled in" using homology modeling or other approaches. Is this mandatory? If so it needs to be clearly communicated in the paper. The creation of parts of models for which there is no clear experimental data to support that model are questionable.

The short answer is no. While it is true that missing parts in the starting structures were filled in using MODELLER or by copying them from other structures (unrelated to both refinement and validation), this is generally not required for the method’s application. The filled-in parts in the starting structures can be split into two groups: (a) parts that are missing in the starting structures and reflected in the density in which case they are allowed to feel the correlation biasing potential, and (b) missing parts that are not reflected in the density and, hence, are subjected only to the atomistic force field. The latter were added for completeness but were later excluded in all subsequent analyses.

A new paragraph at the end of the section “Preparing maps and starting structures” in Materials and methods now clearly communicates whether or not starting structures need to be complete prior to refinement with our method.

11) The authors make an interesting argument about clashes and side- and mainchain outliers. However, arguing that a very low clash score and a significant number of geometry outliers is reasonable stretches credulity. It is well known from high resolution structures what the expectations are for outliers. It is also observed that at lower resolution it becomes increasingly hard to determine correct rotamer and mainchain conformations based on the experimental data. The authors might be better off to emphasize that they are able to create models with a quality consistent with many other methods at this resolution, often better than the starting models, and accept that there is room for improvement.

Considering the discussions in Critical point (2) and point (7), we fully agree with this statement and have rewritten the last paragraph of the section “Force field influence and model accuracy” in the Results part accordingly.

12) The authors should be clear what they mean when they use the phrase "overfitting" in this context. There is often confusion about this term's meaning – e.g. fitting of noise, model correctness, determination of the optimal protocol.

We agree and now provide a statement at the beginning of the subsection “Refinement” about how the term “overfitting” should be understood in the structure refinement context. In fact, the concept of overfitting validation is already incorporated in Figure 2—figure supplement 1 describing our refinement protocol.

13) It would seem appropriate to mention the work of Sanbonmatsu and colleagues: Kirmizialtin et al., 2015.

The work is cited and acknowledged in the Introduction.

[Editors' note: further revisions were requested prior to acceptance, as described below.]

1. Please provide the CABLAM and other statistics (provided in the supplemental files as. tex files) as proper tables.

We were previously asked by the *eLife* editorial support to provide all supplementary tables as separate editable files and include only legends for them in the main article file. For this resubmission, we have compiled the supplementary tables into a single. pdf file and appended it to the end of the main article file (after supplementary figures), as requested. The tables are still available as editable. tex files.

2. You are correct in their assertion that the CDMD procedure has a larger radius of convergence than the Refmac, Phenix and Rosetta procedures. However, while it is very much appreciated that making use of other programs can be challenging, it is probable that some of the protocols used are not optimal for the particular cases being studied. For example, in the case of phenix.real_space_refine the default protocols are appropriate for refinement of models close to correct, i.e. not significantly displaced from the cryo-EM volume (globally or locally).

It is, indeed, true that the tested alternative protocols are not entirely optimal for the presented refinement cases. This is mostly due to the distant starting structures we used to test CDMD. Expectedly, the alternative methods applied to the same distant structures yielded poorer results. Although they were somewhat better than we initially anticipated: for example, the aldolase model generated by Rosetta looked promising, but given the fact that Rosetta was also used to generate the reference aldolase model (5VY3), this result was not surprising. As mentioned in the previous response letters, this was the reason why we refrained from the complete “all-to-all” comparisons in the previous manuscript version. However, we are now convinced that this comparison is necessary and clearly illustrates the challenge of reconciling local refinement accuracy and large convergence radius, in which a proper treatment of concerted molecular motions plays a major role.

Moreover, the protocol outlined in White et al., 2018 helps extend the radius of convergence. However, for models that have more significant starting errors, morphing is an important option for increasing the radius of convergence. It can also be necessary to run more refinement cycles to increase the convergence radius. Please comment in the Discussion.

In all of our Phenix refinements, we switched off the morphing option because we were not expecting the deforming procedure implemented in Phenix (Terwilliger et al., 2012; iterative displacement and smoothing of atom positions in the vicinity each residue to maximize the map-model overlap) to improve the radius of convergence, exactly for the reason mentioned above. To further support this, we have repeated the Phenix refinement of NSF, but this time letting Phenix apply morphing in each macrocycle (this structure is attached to this resubmission; NSF_PHENIX_MORPH.pdb). We have not seen any significant improvements relative to the protocol without morphing in terms of RMSD to the reference structure, and for some solvent-exposed loops, the agreement with the map became even worse.

Concerning the number of refinement cycles, 15 macrocycles used in our Phenix protocols were largely sufficient. In fact, the correlation coefficient (CC_mask_) stopped increasing already after 7th-8th cycle in all of the refinements.

We have added a remark on the above issue in the Conclusion section.

3. The benchmark information in Table 2 is likely to be confusing for many readers (the timings provided in the Benchmark text are very useful). Would it be possible to provide statistics in the form of total run times for refinements in a table form?

Thank you for this suggestion. We agree that providing only the MD simulation performance might be confusing and requires further recalculations to obtain the total run times. We have now added two columns to Table 2 showing the total refinement run times for the short and long protocol versions.